# SORCS1 and SORCS3 control energy balance and orexigenic peptide production

Aygul Subkhangulova[1,*] iD, Anna R Malik[1], Guido Hermey[2], Oliver Popp[1], Gunnar Dittmar[1,3,†],
Thomas Rathjen[1], Matthew N Poy[1], Alexander Stumpf[4], Prateep Sanker Beed[4], Dietmar Schmitz[4],
Tilman Breiderhoff[1] iD & Thomas E Willnow[1,3,**] iD

## Abstract

SORCS1 and SORCS3 are two related sorting receptors expressed in neurons of the arcuate nucleus of the hypothalamus. Using mouse models with individual or dual receptor deficiencies, we document a previously unknown function of these receptors in central control of metabolism. Specifically, SORCS1 and SORCS3 act as intracellular trafficking receptors for tropomyosin-related kinase B to attenuate signaling by brain-derived neurotrophic factor, a potent regulator of energy homeostasis. Loss of the joint action of SORCS1 and SORCS3 in mutant mice results in excessive production of the orexigenic neuropeptide agouti-related peptide and in a state of chronic energy excess characterized by enhanced food intake, decreased locomotor activity, diminished usage of lipids as metabolic fuel, and increased adiposity, albeit at overall reduced body weight. Our findings highlight a novel concept in regulation of the melanocortin system and the role played by trafficking receptors SORCS1 and SORCS3 in this process.

**Keywords** adiposity; agouti-related peptide; brain-derived neurotrophic factor; TrkB; VPS10P domain receptors
**Subject Categories** Metabolism; Neuroscience

## Introduction

VPS10P domain receptors are a unique class of sorting receptors that direct the intracellular transport of target proteins between Golgi, cell surface, and endosomes in mammalian cell types. Sorted cargo includes enzymes, growth factors, and signaling receptors, implicating VPS10P domain receptors in vital cellular functions (reviewed in ref. [1]). Earlier work has largely focused on a role of VPS10P domain receptors in control of protein transport in neurons,

and its relevance for functional integrity but also diseases of the brain, including Alzheimer and Huntington disease, frontotemporal lobar dementia, and schizophrenia (reviewed in ref. [2]). However, genomewide investigations in humans and animal models have also associated VPS10P domain receptors with disorders of the systemic metabolism, including hypercholesterolemia [3], diabetes [4–6], and obesity [7,8], suggesting involvement of these receptors in control of metabolism that warrants further clarification.

Sorting-related receptor CNS expressed (SORCS) 1 exemplifies a member of the VPS10P domain receptor gene family involved in metabolic control [9]. The encoding gene had been associated with type 2 diabetes in mice [4] and with type 1 and type 2 diabetes in humans [5,6]. Subsequent studies identified SORCS1 as a sorting receptor in pancreatic β cells, required to replenish insulin secretory granules. Lack of SORCS1 in gene-targeted mice resulted in impaired insulin secretion from islets when mice were made obese by leptin ablation [10]. Interestingly, SORCS1 shares close homology with another VPS10P domain receptor, termed SORCS3, that has been associated with glucose levels in rats [11]. In fact, 75% identity at the amino acid level and the adjacent localization of both receptor genes in the mammalian genome suggests that they may be the result of a gene duplication event [12]. In contrast to SORCS1, the expression of SORCS3 is restricted to the central nervous system and not seen in the pancreas [13]. Thus, the exact role of SORCS3 in control of metabolism, and its functional interaction with SORCS1, if any, remains unclear.

Here, we have generated novel mouse models with individual or combined defects in *Sorcs1* and *Sorcs3* to shed light on a joint role of both receptors in metabolic control. Individually, both receptor gene defects resulted in increased adiposity in mice that was further aggravated by dual receptor deficiency, supporting the additive action of both receptors in energy homeostasis. Functional studies in mouse and cell models, combined with global proteomics approaches, documented the ability of both receptors to reduce expression of orexigenic neuropeptides, most prominently agouti-related peptide (AgRP), in the arcuate nucleus of the hypothalamus.

1   Max-Delbrueck-Center for Molecular Medicine, Berlin, Germany
2   Institute for Molecular and Cellular Cognition, Center for Molecular Neurobiology, University Medical Center Hamburg-Eppendorf, Hamburg, Germany
3   Berlin Institute of Health, Berlin, Germany
4   Neuroscience Research Center, Charité – University Medicine, Berlin, Germany
    *Corresponding author. Tel: +49 30 9406 3749; E-mail: Aygul.Subkhangulova@mdc-berlin.de
    **Corresponding author. Tel: +49 30 9406 2569; E-mail: willnow@mdc-berlin.de
    †Present address: Department of Oncology, Luxembourg Institute of Health, Strassen, Luxembourg

Because surface exposure and activity of tropomyosin-related kinase B (TrkB), the receptor for brain-derived neurotrophic factor (BDNF) is decreased in SORCS1/3-deficient neurons, we propose that aberrant TrkB signaling in hypothalamic neurons causes a chronic increase in AgRP expression, which, in turn, results in the elevated food intake and defective nutrient partitioning seen in the mutant mice.

## Results

The genes encoding SORCS1 and SORCS3 are closely linked on mouse chromosome 19 (Mouse Genome Informatics: 1929666). To generate mice doubly deficient for both receptors, we made use of a murine ES cell line heterozygous for a floxed *Sorcs3* allele (*Sorcs3*$^{lox/+}$). We had generated this ES cell line previously to produce SORCS3-deficient mice (referred to as S3 KO herein) [14]. *Sorcs3*$^{lox/+}$ ES cells were transfected with a targeting construct to delete exon 1 of the *Sorcs1* locus through homologous recombination (Appendix Fig S1A). ES cell clones carrying both targeted alleles on the same chromosome 19 (*Sorcs1*$^{+/-}$, *Sorcs3*$^{lox/+}$) were used to generate mice doubly deficient for *Sorcs1* and *Sorcs3*, referred to as S1/3 KO (Appendix Fig S1B). From the same targeting experiment, ES cell clones carrying the targeted *Sorcs1* allele but being wild type (WT) for *Sorcs3* (*Sorcs1*$^{+/-}$, *Sorcs3*$^{+/+}$) were used to derive the single SORCS1-deficient mouse line (S1 KO). The breeding strategy to generate all three mutant strains is detailed in the method section. Successful gene inactivation was confirmed by quantitative (q) RT–PCR documenting complete absence of transcripts from the targeted *Sorcs1* and *Sorcs3* alleles in brain tissue of S1/3 KO animals (Appendix Fig S1C). The availability of antibodies directed against mouse SORCS3 enabled us to also document absence of this receptor from brain tissue by Western blot analysis (Appendix Fig S1D).

S1/3 KO mice were born at the expected Mendelian ratio and were viable and fertile. While having normal body weight at birth, the mutant mice weighed less at weaning and throughout adulthood (Fig 1A). The reduced body weight was likely due to a decrease in lean (fat-free) mass as shown by NMR analysis of body composition at 20 weeks of age (Fig 1B). The decrease in lean mass was accompanied by a relative increase in fat mass (Fig 1B). Despite the reduced body weight, S1/3 KO mice displayed an increase in weight of subcutaneous and perigonadal white adipose tissue (WAT) depots (Fig 1C), accompanied by WAT hypertrophy (Fig EV1A and B). In line with increased adiposity, plasma levels of leptin were elevated almost twofold in S1/3 KO mice compared to WT littermates at 18 weeks of age (Fig 1D). The redistribution between fat and lean tissues was also observed in the single S1 KO and S3 KO lines, but was less pronounced than in double-mutant animals, arguing for an additive effect of both gene defects on body composition (Fig 1E). Importantly, the increased adiposity in S1/3 KO mice was evidenced as early as 6 weeks of age (Fig EV1C), although the WAT was not hypertrophic at this young age (Fig EV1D).

Because of the aggravated phenotype seen in the double-mutant as compared to the single-mutant lines (Fig 1E), we focused further analyses on mice lacking both SORCS1 and SORCS3. Indirect gas calorimetry was used to determine basic metabolic rates in these animals at 21 weeks of age. Respiratory exchange ratio (RER), the

ratio of $VCO_2/VO_2$, was higher in S1/3 KO as compared to WT mice, indicating a decrease in relative lipid consumption in mutants (Fig 2A and B). As with WT mice, S1/3 KO animals showed diurnal oscillations in RER, but RER values were increased in the mutants both during the light and the dark cycle. Overall energy expenditure adjusted for lean body mass was not affected by SORCS1/3 deficiency (Fig 2C and D), but the cumulative food intake was chronically increased in S1/3 KO mice as compared to WT animals (Fig 2E). Additionally, the spontaneous locomotor activity was reduced (Fig 2F). The reduction in lipid consumption, as evidenced by increased RER, was not due to an inherent defect in lipolysis in WAT as lipolytic activity in perigonadal adipose tissue explants was unchanged compared to WT tissue as determined by release of glycerol (Fig 2G).

Given the genetic association of *SORCS1* with diabetes and the recently documented role for this receptor in insulin secretion, we also analyzed the systemic glucose metabolism in S1/3 KO mice fed a normal chow. Fasting plasma glucose and insulin levels were unchanged in mutant mice (Fig EV2A). S1/3 KO mice showed a reduced glucose tolerance when challenged with a bolus of glucose in a glucose tolerance test (Fig EV2B and C), but glucose-stimulated insulin secretion (Fig EV2D) and insulin sensitivity (Fig EV2E) were not compromised by S1/3 gene deficiencies. Also, hepatic glucose production, as assessed by pyruvate tolerance test, was normal (Fig EV2F and G). Lastly, the determination of plasma or urine levels of various hormones did not reveal discernible changes in pituitary and adrenal activities in the mutant mice (Table 1).

Because aging aggravates metabolic dysfunctions associated with glucose handling and fat deposition, we explored the consequences of SORCS1/3 deficiencies in an independent cohort of mice at 9–10 months of age. As at younger age, aged S1/3 KO mice displayed elevated RER (Fig 3A and B), increased cumulative food intake (Fig 3C), and reduced locomotor activity (Fig 3D). Remarkably, the genotype-dependent differences in RER and locomotor activity were markedly pronounced in aged mice as compared to 21-week-old animal with, for example, an 8% increase in RER observed at 10 months as opposed to 4% increase at 21 weeks of age. Despite the reduction in locomotor activity in the mutants, the overall energy expenditure adjusted for lean body mass was identical between the genotypes at 9 months of age (Fig 3E and F).

Aging of S1/3 KO mice did not result in manifestation of hyperglycemia (Fig EV3A). Interestingly, although basal circulating insulin levels were normal (Fig EV3B), glucose-stimulated increase in plasma insulin was largely blunted in the aged mutant mice (Fig EV3C). Insulin sensitivity was also slightly decreased in S1/3 KO animals (Fig EV3D). However, glucose tolerance was not affected by the gene deficiency in the aged mice (Fig EV3E and F).

Taken together, ablation of SORCS1 and SORCS3 expressions in mice on a normal chow resulted in a distinct metabolic phenotype with a shift in energy substrate preference, diminished usage of lipids as metabolic fuel, and increased adiposity in the absence of classical obesity manifestation. This metabolic phenotype was obvious at 20 weeks of age and significantly aggravated with age. To identify the tissue causing this unique metabolic phenotype, we explored the co-expression of both receptors in brain and peripheral tissues. The joint expression of *Sorcs1* and *Sorcs3* was largely confined to the central nervous system (CNS) with highest transcript

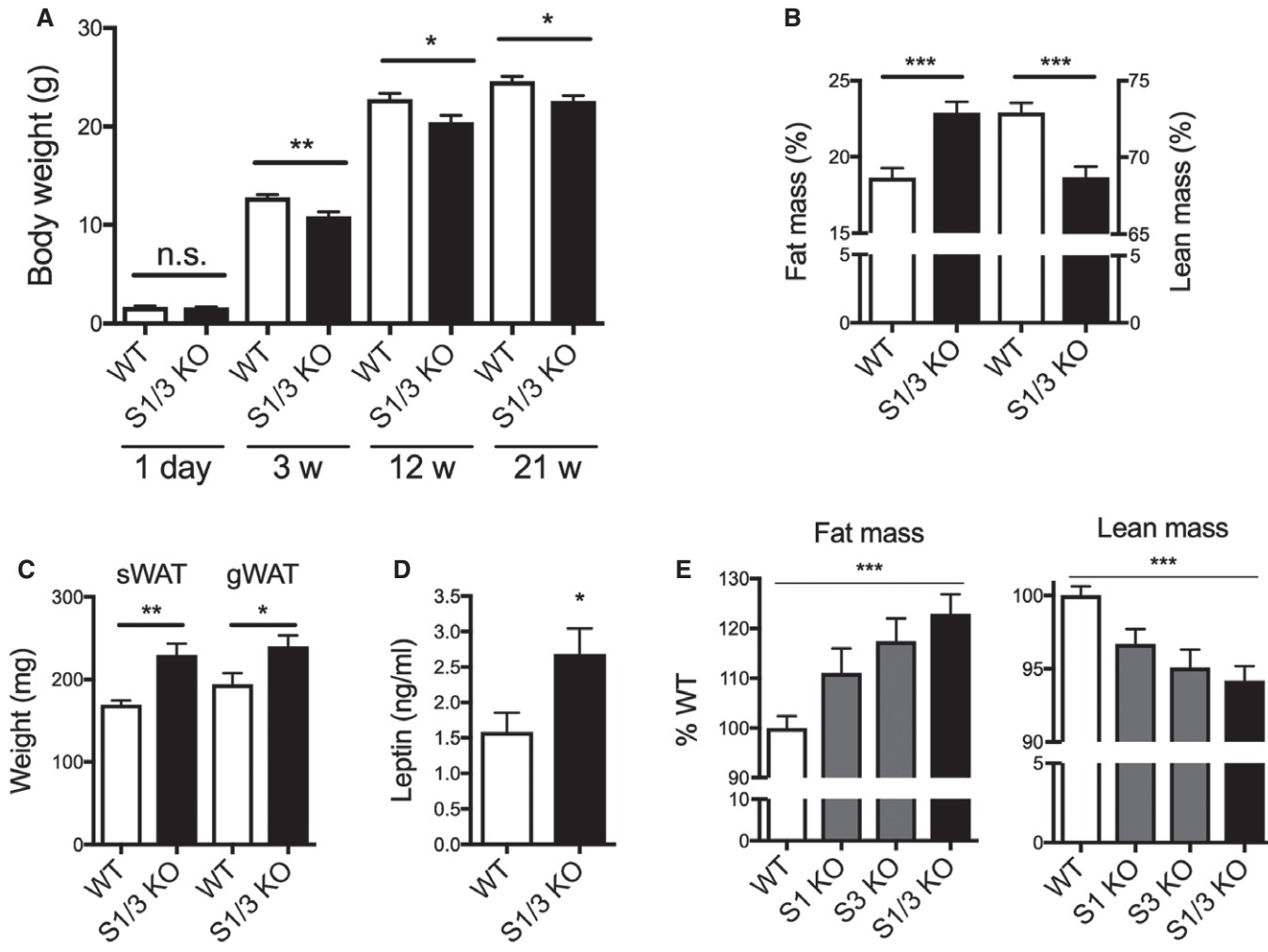

**Figure 1. Altered body composition and increased adiposity in mice with single or combined SORCS1 and SORCS3 deficiencies.**

A   Body weight of WT and S1/3 KO mice at different ages. Decreased body weight in S1/3 KO mice was observed starting from 3 weeks of age but not at post-natal day 1 (n = 7–15 animals/group).

B   Body composition as determined by NMR imaging in 20- to 22-week-old mice of the indicated genotypes. S1/3 KO mice show an increase in fat mass and a concomitant decrease in lean (fat-free) mass as compared to WT controls (n = 12–15 animals/group).

C   Weight of subcutaneous (sWAT) and perigonadal (gWAT) white adipose tissue depots in WT and S1/3 KO mice at 16 weeks of age (n = 5–12 mice/group).

D   Plasma leptin levels after overnight fasting in 18-week-old WT and S1/3 KO (n = 8 mice/group).

E   Determination of fat and lean tissue mass in mice of the indicated genotypes using NMR. Each KO line was compared to the corresponding WT littermates. Percent fat (or lean) mass in WT was set to 100% (n = 5–15 mice/group). Asterisks indicate results of the comparison between all four groups by one-way ANOVA ($P < 0.001$). Comparisons between WT and individual KO lines were performed by Bonferroni's post-test: $P > 0.05$ for S1 KO, $P < 0.05$ for S3 KO, $P < 0.001$ for S1/3 KO.

Data information: All data are shown as mean ± SEM and were analyzed using a two-tailed unpaired t-test, unless otherwise stated (*$P < 0.05$, **$P < 0.01$, ***$P < 0.001$).

---

levels in cortex and hypothalamus and lower levels in hippocampus (Fig 4A and B). With relevance to central control of metabolism, expression of both receptors in the hypothalamus was noteworthy. In this brain region, transcript levels were higher for *Sorcs3* than for *Sorcs1* (Fig 4C). *Sorcs1* transcripts showed a compensatory upregulation in hypothalami lacking *Sorcs3*, whereas levels of *Sorcs3* remained unchanged in the SORCS1-deficient hypothalamus (Fig 4D). Using *in situ* hybridization, expression of *Sorcs1* was detected in the dorsomedial nucleus (DMH), the ventromedial nucleus (VMH), and the arcuate nucleus (Arc) of the hypothalamus. Strong *Sorcs3* expression was seen in VMH and Arc (Fig 4E).

Additionally, *Sorcs3* expression was detected in the paraventricular hypothalamic nucleus (PVN; Appendix Fig S2).

Co-expression of *Sorcs1* and *Sorcs3* in several nuclei of the hypothalamus suggested a defect in hypothalamic circuitry as the underlying cause of the altered energy metabolism in S1/3 KO mice. In support of this hypothesis, transcript levels for the appetite-stimulating neuropeptide agouti-related peptide (AgRP) were increased in mutants compared to WT mice. Increased expression of *Agrp* in S1/3 KO was independent of the feeding status of the mice and seen under fasted and fed conditions (Fig 5A). In mice fed *ad libitum*, mRNA levels of another orexigenic factor, neuropeptide Y (NPY),

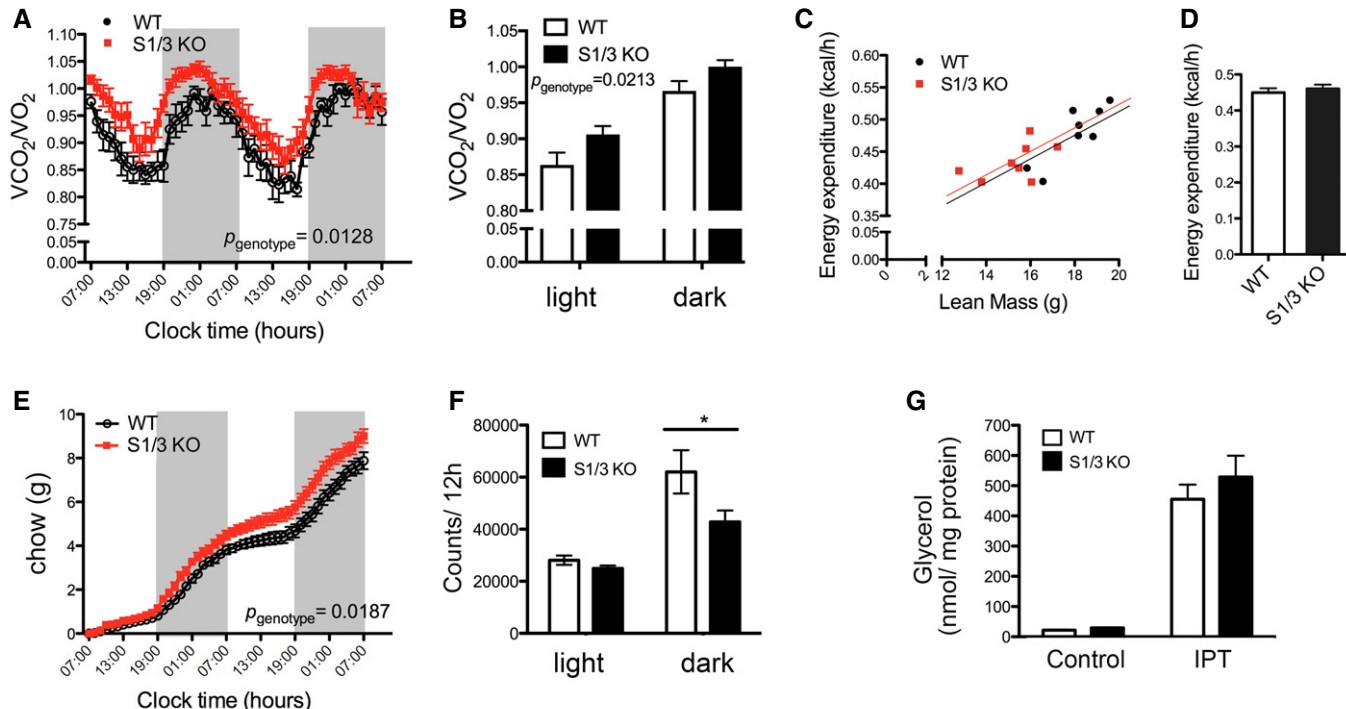

**Figure 2.  Impaired energy homeostasis in S1/3KO mice.**

WT and S1/3 KO mice were subjected to metabolic profiling by indirect calorimetry at 20–22 weeks of age ($n$ = 8 mice/group).

A   Dynamic pattern of respiratory exchange ratio (RER) in WT and S1/3 KO. Elevated RER reflects a decrease in relative lipid metabolism in S1/3 KO animals as compared to littermate controls.

B   Average RER values (from A) during the light and the dark phase (4 days and four nights average, respectively).

C   24-h energy expenditure of individual mice plotted against their lean mass.

D   24-h energy expenditure as analyzed by ANCOVA and adjusted for differences in lean body mass between the genotypes ($P$ = 0.6378).

E   Cumulative food intake in WT and S1/3 KO mice over the course of 2 days and two nights.

F   Spontaneous locomotor activity of mice determined as the number of beam crossings per day and night (averaged for 4 days and four nights, respectively).

G   Lipolytic activity, as determined by glycerol release from perigonadal adipose tissue explants, is not affected by loss of SORCS1/3. The glycerol concentration in the medium was measured after 1-h incubation of tissue explants either in the absence (basal) or in the presence (IPT) of 10 μM isoproterenol ($n$ = 8–11 mice/group).

Data information: In all panels, except for (C), data are shown as mean ± SEM. Data were analyzed using two-way ANOVA with Bonferroni post-test (A, B, E–G) or ANCOVA (C, D). *$P$ < 0.05.

were also elevated, accompanied by a decrease in expression of proopiomelanocortin (POMC), the precursor of the anorexigenic alpha-melanocyte-stimulating hormone (Fig 5A). The aberrant rise in AgRP expression in mutants was observed as early as 8 weeks of age and persisted in aged mice (35 weeks; Fig 5B), suggesting a specific and chronic increase in the number and/or activity of AgRP-producing neurons in S1/3 KO animals.

The elevation in hypothalamic AgRP production was not due to an increase in the number of AgRP neurons as shown by crossing

S1/3 KO mice with a reporter strain expressing GFP under control of the *Npy* promoter (*Npy*-GFP mice) [15]. Counting GFP+ cells in the Arc of (*Npy*-GFP;S1/3 KO) and (*Npy*-GFP; WT) animals, no discernible difference in cell numbers was observed comparing genotypes (Fig 5C and D).

In previous studies, both SORCS1 and SORCS3 were implicated in modulation of synaptic activity through trafficking of ionotropic glutamate receptors [14,16]. Accordingly, we analyzed the consequences of SORCS1/3 deficiency for the excitability of AgRP

**Table 1.  Levels of circulating hormones and metabolites in overnight fasted WT and S1/3 KO mice**

| Hormone | Sample | WT | S1/3 KO | P value |
|---|---|---|---|---|
| Growth hormone (pg/ml) | Plasma | 153.4 ± 15.2 | 179.5 ± 21.0 | 0.3326 |
| Adrenocorticotropic hormone (pg/ml) | Plasma | 251.0 ± 14.1 | 205.9 ± 24.3 | 0.1847 |
| Corticosterone (ng/ml) | Plasma | 131.0 ± 24.3 | 170.5 ± 34.4 | 0.3460 |
| Epinephrine (μg/g creatinine) | Urine | 90.1 ± 9.1 | 108.0 ± 9.0 | 0.1785 |
| Norepinephrine (μg/g creatinine) | Urine | 500.5 ± 33.8 | 541.1 ± 25.0 | 0.3392 |

Data are shown as mean ± SEM and were analyzed using a two-tailed unpaired $t$-test ($n$ = 5–15 mice/group).

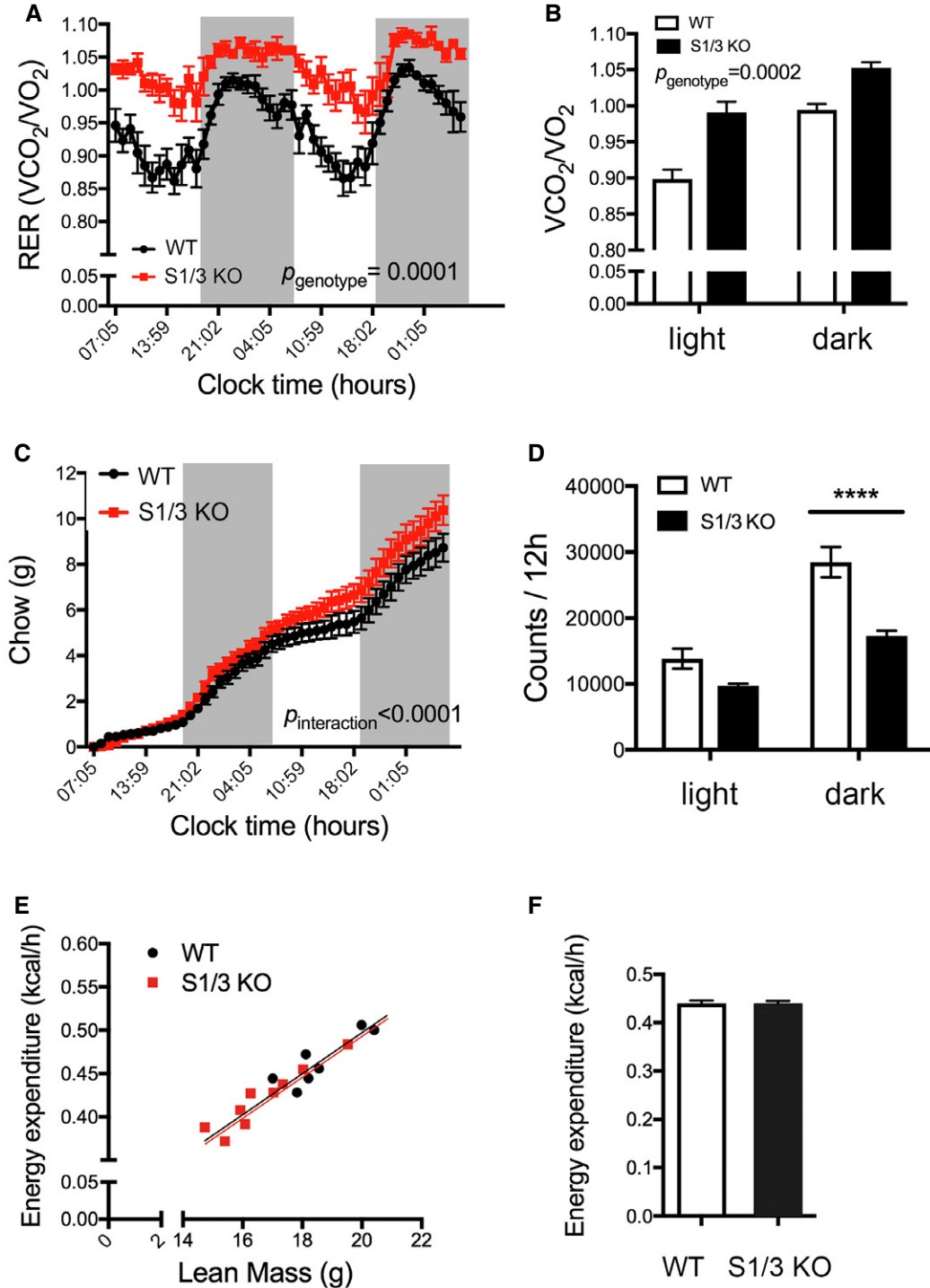

**Figure 3.  Impaired energy homeostasis in aged S1/3KO mice.**

WT and S1/3 KO mice were subjected to metabolic profiling by indirect calorimetry at 9–10 months of age (n = 7–8 mice/group).

A  Dynamic pattern of respiratory exchange ratio (RER) in WT and S1/3 KO.
B  RER values (from A) during the light and the dark phase (3 days and three nights average, respectively).
C  Cumulative food intake in WT and S1/3 KO mice over the course of 2 days and two nights.
D  Spontaneous locomotor activity of mice determined as the number of beam crossings per day and per night (averaged for 3 days and three nights, respectively).
E  24-h energy expenditure of individual mice plotted against their lean mass.
F  24-h energy expenditure as analyzed by ANCOVA and adjusted for differences in lean body mass between the genotypes (P = 0.9045).

Data information: In all panels, except for (E), data are shown as mean ± SEM. Data were analyzed using two-way ANOVA with Bonferroni post-test (A–D) or ANCOVA (E, F). ****$P < 0.0001$.

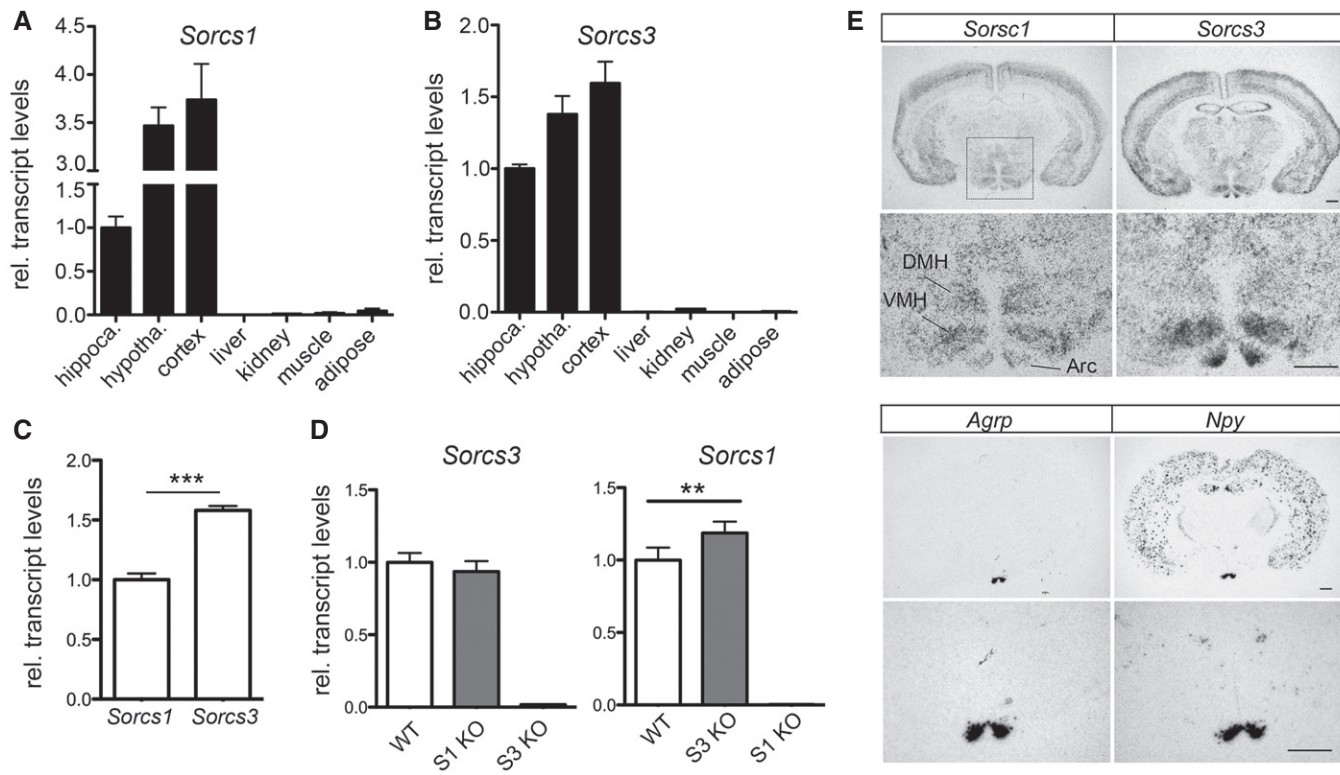

**Figure 4. Co-expression of *Sorcs1* and *Sorcs3* in the hypothalamus.**

A, B    Transcript levels for *Sorcs1* (A) and *Sorcs3* (B) in the indicated mouse tissues were assessed by quantitative (q) RT–PCR. Expression levels in the hippocampus were set to 1 (*n* = 3 mice/group, 10 weeks of age).

C    Comparison of *Sorcs1* and *Sorcs3* transcript levels using qRT–PCR in hypothalami of WT mice. Identical amplification efficiency for both gene expression assays was validated in a separate experiment (*n* = 5 mice/group).

D    Transcript levels for *Sorcs1* and *Sorcs3* as assessed by qRT–PCR in hypothalami from mice with single *Sorcs1* (S1 KO) or *Sorcs3* (S3 KO) deficiencies. S3 KO mice show a compensatory increase in *Sorcs1* expression compared to WT controls. Expression in WT was set to 1 (*n* = 5–10 mice/group).

E    *In situ* hybridization (ISH) for *Sorcs1* and *Sorcs3* on coronal brain sections indicating expression of both receptors in cerebral cortex and in various nuclei of the hypothalamus (Arc: arcuate nucleus; VMH: ventromedial nucleus; DMH: dorsomedial nucleus). ISH for *Agrp* and *Npy* on adjacent sections was used as controls for identification of the arcuate nucleus. For each gene, the lower panel represents a higher magnification of the hypothalamus area (marked in the overview micrograph of ISH for *Sorcs1*). Scale bar: 500 μm; *n* = 3 mice.

Data information: Data in (A–D) are shown as mean ± SD and were analyzed using a two-tailed unpaired *t*-test. **P < 0.01, ***P < 0.001.

neurons. To do so, membrane potentials and firing rates of AgRP neurons were recorded in acute brain slices from fed *Npy*-GFP mice. No genotype-specific difference was observed in firing rates of AgRP neurons (Appendix Fig S3A), although AgRP neurons from S1/3 KO mice were slightly depolarized (Appendix Fig S3B).

To identify the cause for the upregulation of *Agrp* transcription in mutant mice, we assessed the levels of Krüppel-like factor (KLF4), a transcription factor that potently induces *Agrp* gene transcription *in vitro* and *in vivo* [17–19]. In line with aberrant induction of *Agrp* transcription under fed conditions, levels of KLF4 in the Arc were significantly increased in S1/3 KO mice as documented by quantitative immunohistochemistry (Fig 5E and F).

AgRP is one of the mediators of AgRP/NPY neuronal action in metabolic control. It acts as an inverse agonist of melanocortin-3 and melanocortin-4 receptors (MC3R, MC4R) to stimulate feeding and to decrease energy expenditure in a long-lasting manner [20–22]. Both *Agrp* expression and AgRP/NPY neuronal activity are inhibited by leptin, in part mediating leptin's anorexigenic effects [23,24]. In line with the chronically increased AgRP levels, leptin

injection caused a lesser reduction of food intake in S1/3 KO mice compared to WT controls (Fig 6A). As another indication of elevated AgRP levels and downregulated melanocortin receptor signaling, the cold tolerance of mutant mice was attenuated, as judged by a stronger decrease in body core temperature in response to short-term cold exposure compared to WT littermates (Fig 6B). In line with the latter defect, the relative expression of thermogenic genes was decreased in the WAT of cold-exposed S1/3 KO mice compared to controls (Fig 6C).

Multiple mechanisms may be envisioned how sorting receptors SORCS1 and SORCS3 may, directly or indirectly, control AgRP expression in hypothalamic neurons. To explore the underlying molecular concept, we turned to an unbiased proteomics approach comparing the surface proteome of WT and S1/3 KO primary neurons. We reasoned that combined absence of these sorting receptors should result in aberrant distribution of yet unknown cargo between cell surface and intracellular compartments. To identify such targets, surface proteins in primary cortical neurons from WT and S1/S3 KO animals were biotinylated. Subsequently,

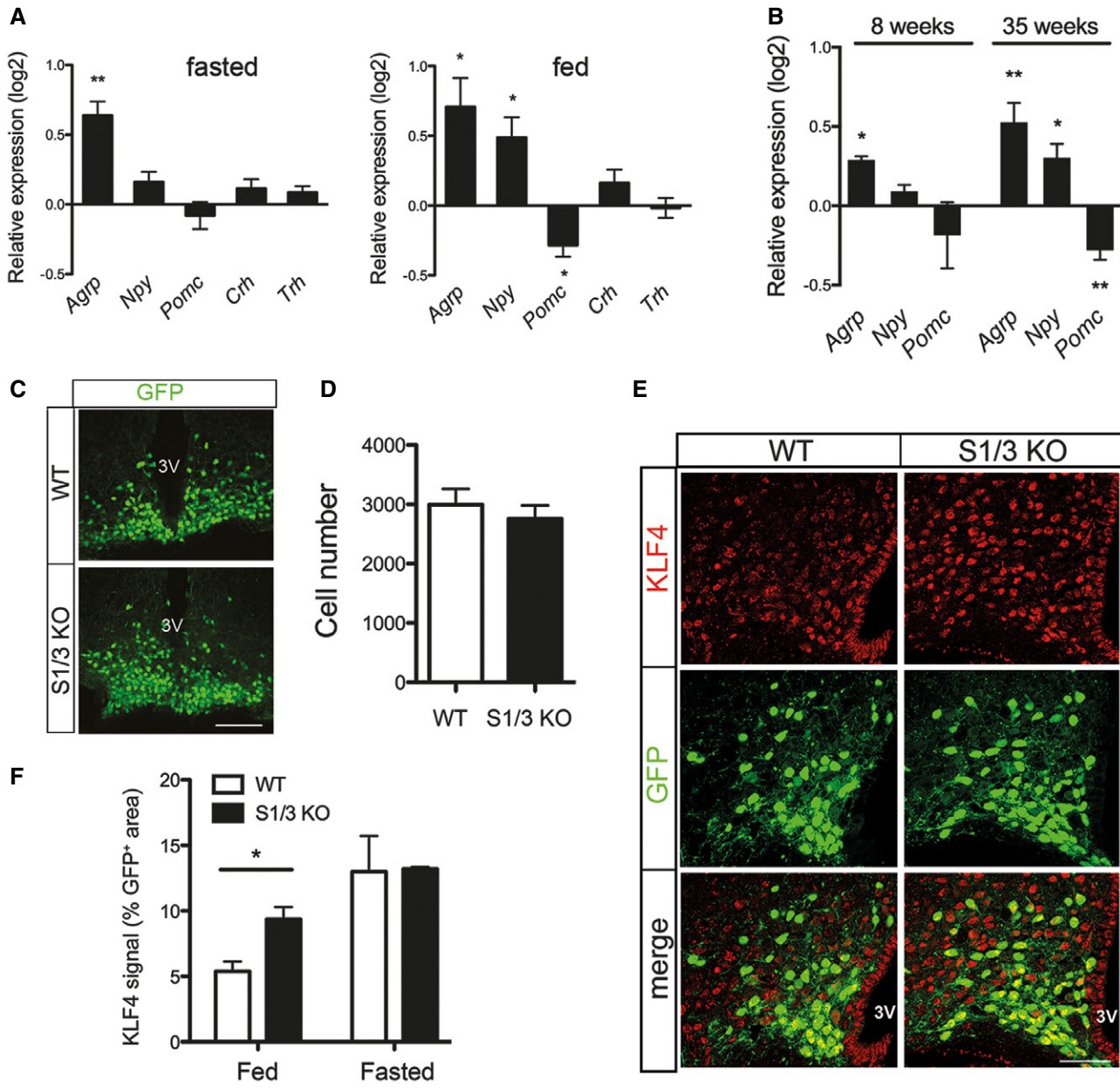

**Figure 5. Loss of SORCS1 and SORCS3 increases hypothalamic expression of *Agrp* and *Klf4*.**

A   Expression of hypothalamic neuropeptides and hormones as assessed by quantitative (q)RT–PCR in overnight fasted and *ad libitum* fed mice at 21 weeks of age. Log2-fold change expression in S1/3 KO relative to the expression in WT is shown. *Agrp* transcript levels are increased in S1/3 KO under both fasted and fed conditions. *n* = 8–11 (fasted) or 4–5 (fed) mice/group. *Agrp*, agouti-related peptide; *Npy*, neuropeptide Y; *Pomc*, proopiomelanocortin; *Crh*, corticotropin-releasing hormone; *Trh*, thyrotropin-releasing hormone.

B   Expression of hypothalamic neuropeptides and hormones as assessed by quantitative (q) RT–PCR in overnight fasted mice at 8 or 35 weeks of age. Log2-fold change expression in S1/3 KO relative to the expression in WT is shown. *n* = 4–6 (8-week-old) or 14–16 (35-week-old) mice/group.

C   Detection of NPY/AgRP neurons by native fluorescence of GFP (green) in the arcuate nucleus of *Npy*-hrGFP mice either wild type (*Npy*-GFP/WT) or homozygous deficient for SORCS1/3 (*Npy*-GFP/S1/3 KO) at 10 weeks of age. Mice were fed *ad libitum*. Scale bar: 100 μm; 3V, third ventricle.

D   The total number of NPY/AgRP neurons, as identified by GFP fluorescence, is not affected by loss of SORCS1/3. Six histological sections across the arcuate nucleus per *Npy*-GFP/WT or *Npy*-GFP/S1/3 KO mouse (as exemplified in panel C) were scored (*n* = 3 mice/group).

E   Immunodetection of KLF4 (red) in the arcuate nucleus of *Npy*-GFP/WT and *Npy*-GFP/S1/3 KO mice. NPY/AgRP neurons were identified by GFP fluorescence (green). Mice were fed *ad libitum* (*n* = 6 mice/group). Scale bar: 100 μm; 3V, third ventricle.

F   Quantification of the KLF4-positive area in NPY/AgRP neurons on histological sections as exemplified in (E). Two sections per mouse in the middle part of the arcuate nucleus (the region 1.82–1.94 mm caudal to bregma) were analyzed. *n* = 3 (fasted) or 6 (fed) mice/group.

Data information: Data are shown as mean ± SEM and were analyzed using a two-tailed unpaired *t*-test (A, B, D) or two-way ANOVA with Bonferroni post-test (F). *P* < 0.05, **P* < 0.01.

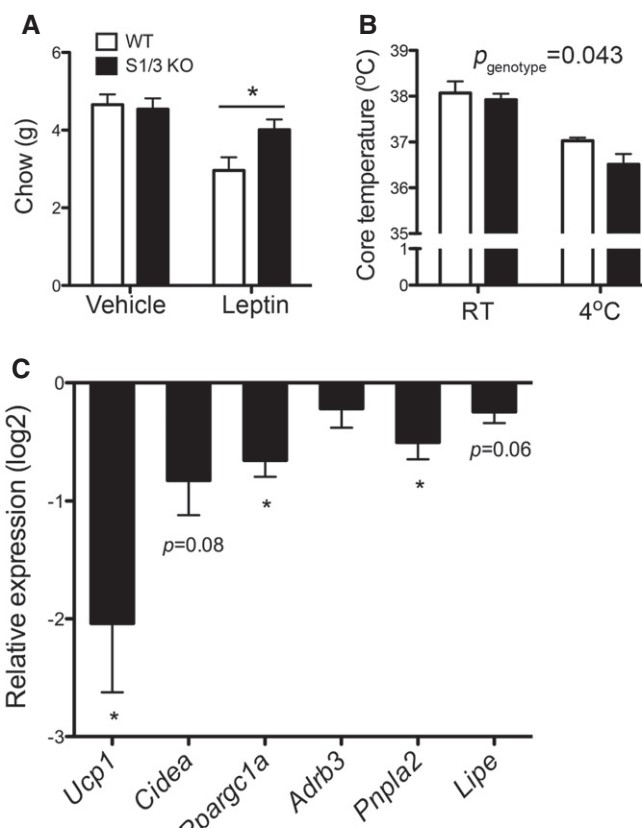

**Figure 6. Pathophysiological effects of dysregulated melanocortin system in S1/3KO mice.**

A   The effect of peripheral leptin (2.5 mg/kg body weight) or vehicle (phosphate-buffered saline, PBS) administration on food intake in 15-week-old mice measured over a 24-h period (n = 8 mice/group). The leptin sensitivity was reduced in S1/3 KO mice.

B   Rectal temperature in 16-week-old mice of the indicated genotypes kept at room temperature and after 6 h of cold exposure (4°C) (n = 7–8 mice/group). Cold-induced thermogenesis is impaired in S1/3 KO mice (two-way ANOVA, $p_{genotype}$ = 0.0435).

C   Expression of thermogenic genes in perigonadal adipose tissue from mice after 6-to 16-h cold exposure (4°C). Quantitative RT–PCR data are shown as log2-fold change expression in S1/3 KO relative to the expression in WT (n = 13–14 mice/group). Cold-induced browning of white adipose tissue is impaired in S1/3 KO as judged by decreased expression of the indicated genes. *Ucp1*, uncoupling protein 1; *Cidea*, cell death activator CIDE-A; *Pppargc1a*, peroxisome proliferator-activated receptor γ co-activator 1-α; *Adrb3*, β-3 adrenergic receptor; *Pnpla2*, adipose triglyceride lipase; *Lipe*, hormone-sensitive lipase.

Data information: Data are shown as mean ± SEM and analyzed using a two-tailed unpaired *t*-test (C) or two-way ANOVA with Bonferroni post-test (A, B). *P < 0.05.

biotinylated proteins were purified from cell extracts using strepta-vidin beads and subjected to quantitative label-free proteomics using LC-MS/MS (Fig 7A). Several proteins with altered cell surface exposure in S1/3 KO as compared to control neurons were identified, including receptors, transporters as well as proteins involved in vesicle trafficking and neurotransmitter release (Fig 7B, Table EV1). Proteins with altered surface exposure in S1/3 KO neurons included the established SORCS1 ligands amyloid precursor protein [25] and neurexin [16], providing proof for the feasibility of

our approach to identify novel receptor targets (Table EV1). One candidate with particular relevance to VPS10P domain receptor biology and to metabolic control, that caught our attention, was TrkB, encoded by *Ntrk2*. Surface exposure of TrkB was increased 1.57-fold in S1/3 KO neurons as compared to controls (*P* = 0.02; Table EV1). TrkB acts as high-affinity receptor for BDNF, a regulator of neuronal survival, differentiation, and synaptic plasticity (reviewed in ref. [26]). With relevance to this study, perturbations of the BDNF signaling pathway also result in metabolic imbalance in mice and humans, but the mechanisms whereby this neurotrophin affects satiety and energy expenditure are still not fully understood (reviewed in refs. [27,28]).

Altered subcellular localization of TrkB in neurons from S1/3 KO mice was further substantiated *in vivo* by subjecting mouse brains to membrane fractionation (Fig 7C) and subsequent Western blot analysis (Fig 7D). The accuracy of the membrane fractionation was confirmed by the enrichment of protein markers (PSD95, synapto-physin, Golgin 97) for various membrane compartments in these fractions (Fig 7D). Quantitative analysis of replicate experiments showed that loss of SORCS1 and SORCS3 resulted in the redistribution of TrkB between the light membrane-enriched (P3) and synaptosomal membrane (P2) fractions (Fig 7E). Whereas TrkB levels were equally distributed between P3- and P2-fractions in WT brains, the receptor pool was shifted from synaptosomal- toward light membrane-enriched fraction in S1/3 KO brains, suggesting accumulation of TrkB in sorting vesicles.

Intriguingly, VPS10P domain receptors sortilin [29], SORLA [30], and SORCS2 [31] have been shown to interact with TrkB and to facilitate functional expression of this receptor. Sortilin and SORLA facilitate trafficking of TrkB in neurites of dorsal root ganglion and hippocampal neurons, respectively. SORCS2 targets TrkB to the post-synaptic density of hippocampal neurons. Using co-immuno-precipitation from transiently transfected Chinese hamster ovary cells, we now document the ability of SORCS1 and SORCS3 to also bind TrkB (Fig 7F and G).

At the molecular level, BDNF binding causes dimerization and transphosphorylation of TrkB, triggering activation of several signaling pathways, ultimately leading to the induction of BDNF target gene transcription. When treated with exogenously added BDNF, S1/3 KO cortical neurons showed an acutely increased response in terms of TrkB phosphorylation (Fig 8A and B). This observation suggested an inhibitory action of SORCS1 and SORCS3 on TrkB that contrasts the stimulatory activity seen for all other VPS10P domain receptors. This inhibitory action of SORCS1 and SORCS3 was also apparent in acute hypothalamic slices with elevated levels of phosphorylated (p) TrkB seen in S1/3 KO as compared to WT tissues following BDNF stimulation (Fig 8C and D).

To verify that the documented interaction of SORCS1 and SORCS3 with TrkB can occur in AgRP neurons, we tested expression of both receptors and TrkB in this neuronal population. To do so, we performed fluorescence-activated cell sorting (FACS) on dissociated arcuate nuclei from adult *Npy*-GFP mice (Fig 8E). In sorted GFP[+]-neurons, *Agrp* transcripts were strongly enriched, whereas transcripts for growth hormone-releasing hormone (*Ghrh*), which is not expressed by AgRP neurons [32], were undetected, indicating specific isolation of the AgRP neuronal transcriptome in our assay. Expression of both *Sorcs1* and *Sorcs3* was detected in AgRP neurons, with *Sorcs3* transcripts being 2.5-fold enriched in this neuronal population as

compared to the transcriptome of the unsorted arcuate nucleus. Expression of *Ntrk2* (TrkB) was also clearly detected in AgRP neurons (Fig 8E), in line with findings from single-cell transcriptomics studies performed on dissociated AgRP neurons by others [33].

Next, we asked whether BDNF could affect *Agrp* expression through the action of KLF4 in hypothalamic neurons. Treatment of primary hypothalamic neurons with BDNF for 48 h caused a strong induction in *Agrp* expression (Fig 8F). Along with *Agrp*, expression of *Npy* and somatostatin (*Sst*) was also increased in BDNF-treated hypothalamic neurons. The two latter genes were reported to be responsive to BDNF in cultured cortical neurons and in various brain areas *in vivo* [34–37]. In contrast, transcript levels of *Ghrh* were not affected by the neurotrophin treatment, documenting that upregulation of *Agrp* and *Npy* expression was not due to a general augmentation of neuronal viability by BDNF. Though the exact pathway for BDNF-dependent induction of *Agrp* transcription is unclear yet, KLF4 is a plausible candidate for mediating this effect as it is a target of neurotrophin signaling in established cell lines and primary hippocampal neurons [38–40]. In line with this notion, *Klf4* expression was rapidly upregulated by BDNF in the primary hypothalamic neurons in our experiments (Fig 8G).

Based on the altered subcellular localization of TrkB in the mutant brain (Fig 7D and E), SORCS1 and SORCS3 may regulate axonal and/or dendritic transport of TrkB-containing vesicles in neurons, thereby affecting surface exposure of the receptor and its availability to BDNF. To substantiate this model, we analyzed trafficking of a green fluorescent protein (GFP)-tagged TrkB in live primary cortical neurons from WT and S1/3 KO mice. Analysis of kymographs visualizing axonal transport of GFP-TrkB did not reveal any significant difference in the kinetics parameters of TrkB-containing vesicles comparing WT and S1/3 KO neurons (Fig EV4). However, the distribution of retrogradely (from neurite toward cell body) and anterogradely (from cell body toward neurite) moving cargo vesicles tended to be affected by the genotype, with possibly decreased retrograde and increased anterograde pools in S1/3 KO neurons (Fig EV4C). In line with this observation, the total distance (run length) covered by TrkB vesicles within an uninterrupted period of retrograde movement showed a tendency to being decreased in S1/3 KO neurons (Fig EV4D).

Our data are consistent with a model whereby SORCS1 and SORCS3 functionally interact with TrkB to alter its subcellular localization and downregulate signaling by BDNF in hypothalamic neurons. Absence of these sorting receptors increases the bioactive TrkB pool at the neuronal cell surface and coincides with chronic transcriptional induction of *Agrp* through the BDNF target KLF4, ultimately impairing energy homeostasis in S1/3 KO mice.

## Discussion

We applied novel mouse models with individual or combined deficiencies for SORCS1 and SORCS3 to explore roles for these related receptors in control of metabolism. Combined loss of SORCS1 and SORCS3 causes a unique metabolic phenotype that impacts body composition with accumulation of fat over lean tissue mass. Altered body composition is accompanied by a shift in metabolic fuel preference with preferred usage of carbohydrates over lipids. In addition to defective nutrient partitioning, S1/3 KO mice display increased cumulative food intake and decreased locomotor activity, resulting in a state of chronic energy excess. This positive energy balance likely explains the increased adiposity seen in mice with individual and, in aggravated form, in animals with combined receptor gene defects.

No major alterations in glucose homeostasis were detected in S1/3 KO mice on a normal chow. Though the mutant mice showed a mild impairment in glucose tolerance at young age (12 weeks; Fig EV2B and C), this phenotype did not progress to hyperglycemia during aging (Fig EV3A), suggesting that SORCS1 and SORCS3 are dispensable for glycemic control in chow-fed mice. Aged S1/3 KO animals exhibit a reduction in insulin sensitivity as well as decreased insulin levels after a bolus of glucose (Fig EV3C and D), consistent with the phenotype documented for mice singly deficient for SORCS1 [10]. Importantly, SORCS1-deficient mice developed these defects exclusively when made genetically obese by introduction of the *leptin^{ob}* mutation, while our model displayed blunted insulin secretion with advanced age only. Jointly, these findings support the notion that type 2 diabetes associated *Sorcs1* plays a role in complex gene–environment interactions in this disease. Defects in glucose handling in obese *Sorcs1* mutants were attributed to the loss of SORCS1 from pancreatic islets, where SORCS3 is not expressed [4]. Our finding that combined ablation of SORCS1 and

---

**Figure 7.  Altered subcellular localization of TrkB in primary neurons and brains from S1/3 KO mice.**

A    Workflow of the cell surface proteome analysis in primary neurons.

B    Results of quantitative label-free proteomics comparing the surface proteomes of WT and S1/3 KO primary cortical neurons. Plot represents −log10(*P*-value) and log2 (relative levels S1/3KO/WT) obtained for each protein. For proteins that were detected in either S1/3 KO or WT samples only, the log2(S1/3 KO/WT) was set to 10 or -10, respectively. Threshold values are 1.3 for −log10(*P*-value) and ±0.3 for log2(S1/3KO/WT). Proteins showing higher (green) or lower (red) surface levels in S1/3 KO as compared to WT neurons are indicated. Proteins with unchanged cell surface levels are indicated in blue. *n* = 3 biological replicates/group (with each biological replicate run in two technical replicates).

C    Scheme of brain membrane fractionation used to analyze subcellular localization of TrkB *in vivo* (see method section for details).

D    Representative Western blotting of TrkB in various brain membrane fractions obtained as described in panel (C). Highlighted lanes represent P3 and P2 fractions, quantified in (E). Detection of PSD95, synaptophysin, and Golgin 97 was used to assess the accuracy of subcellular fractionation.

E    Quantification of total TrkB distribution between P3 (light membrane fraction) and P2 (synaptosomal fraction) from Western blots exemplified in panel (D) (*n* = 5 mice/group, 7–12 weeks of age). The TrkB pool is shifted toward the light membrane (P3) fraction in S1/3 KO brains. Data are shown as mean ± SEM and analyzed using two-way ANOVA with Bonferroni post-test. ****$P < 0.0001$.

F, G    Co-immunoprecipitation of SORCS1 and SORCS3 with TrkB. TrkB was immunoprecipitated from transiently transfected Chinese hamster ovary cells and the presence of SORCS1 or SORCS3 in the immunoprecipitates documented by Western blotting. Panel input shows levels of the indicated proteins in the cell lysates prior to immunoprecipitation. Panel IP documents co-immunoprecipitation of SORCS1 (F) and SORCS3 (G) from cells expressing (lanes 3) but not from cells lacking TrkB (lanes 2). The experiment was replicated twice.

Source data are available online for this figure.

SORCS3 impairs insulin secretion in lean mice, argues that centrally expressed SORCS3 contributes to the regulation of insulin levels. In this respect, expression of SORCS3 in the hypothalamus is of particular interest, since hypothalamic neurons sense glucose and control secretion of pancreatic hormones [41].

The hypothalamus is also one of the key brain areas in control of food intake and nutrient partitioning. Ablation of SORCS1 and SORCS3 results in the increased expression of AgRP (Fig 5A and B),

an orexigenic neuropeptide expressed mainly by Arc neurons. Gain of AgRP function achieved experimentally by injection or over-expression of the peptide evokes phenotypes that are seen chronically in S1/3 KO mice, including elevated food intake, increased RER, decreased locomotor activity, reduced body temperature, and adipose tissue accumulation and hypertrophy [22,42–44]. *Sorcs1* and *Sorcs3* are expressed in AgRP neurons (Fig 8E), with *Sorcs1* expression being profoundly regulated by fasting [33]. These data

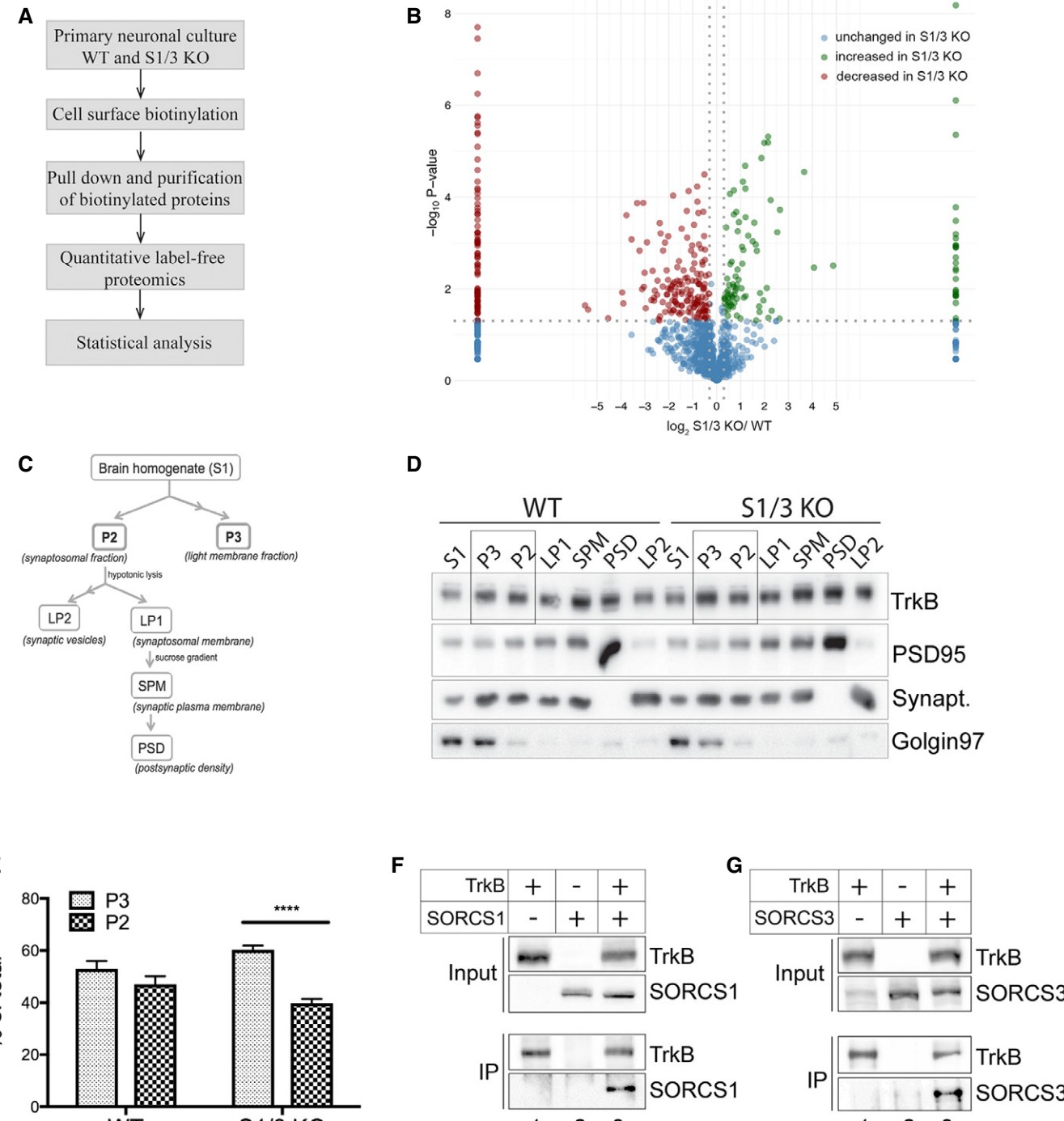

**Figure 7.**

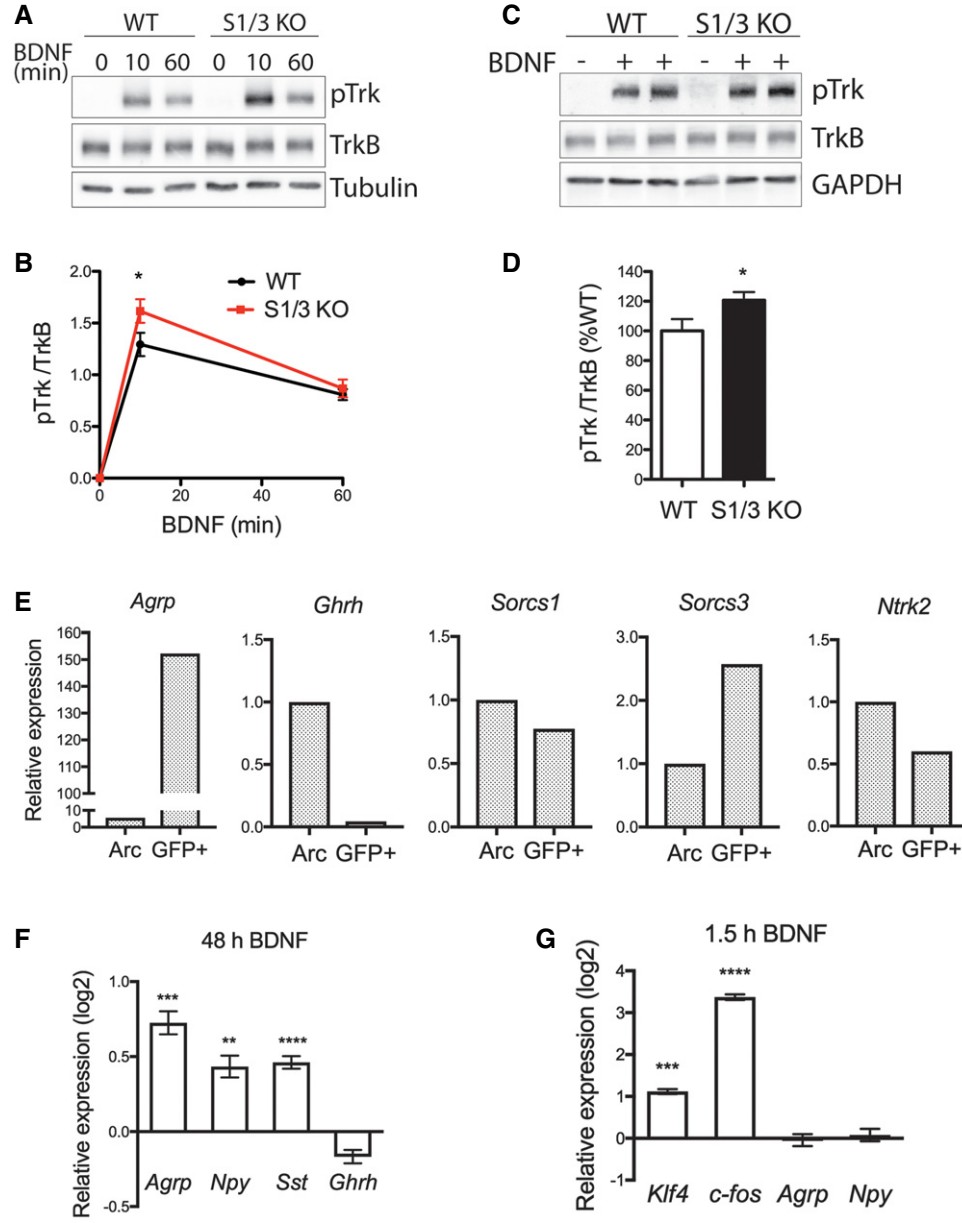

**Figure 8.  Deficiency for SORCS1 and SORCS3 enhances BDNF signaling in primary cortical neurons and acute hypothalamic slices.**

A  Representative Western blots showing phosphorylated TrkB (pTrk) and total TrkB levels in DIV7 primary cortical neurons treated with BDNF (100 ng/ml) for the indicated periods of time. Detection of tubulin served as loading control.

B  Quantification by densitometric scanning of replicate blots (n = 4 biological replicates/genotype), as exemplified in panel (A), documents an increase in pTrkB levels in S1/3 KO neurons compared to WT neurons 10 min after BNDF application.

C  Representative Western blots showing phosphorylated TrkB (pTrk) and total TrkB levels in acute hypothalamic slices of the indicated genotypes treated with BDNF (200 ng/ml) for 1 h. Detection of GAPDH served as a loading control.

D  Quantification by densitometric scanning of replicate blots (n = 14–15 mice/genotype), as exemplified in panel (C), documents an increase in pTrkB levels in S1/3 KO compared to WT hypothalamic slices.

E  Expression of the indicated genes as analyzed by qRT–PCR in FACS-sorted NPY/AgRP neurons (GFP[+]) from fasted *Npy*-GFP mice. Expression in the sorted neurons was related to the expression in the whole (dissociated, but not FACS-sorted) arcuate nucleus (Arc). Tissue from 10 mice was pooled for the experiment.

F  Expression of neuropeptides as analyzed by qRT–PCR in DIV8 primary hypothalamic neurons treated with BDNF (100 ng/ml) for 48 h. The log2-fold change in expression relative to the expression levels in vehicle-treated neurons is shown. n = 5–6 technical replicates/treatment; data are from two neuronal culture preparations; 8-9 hypothalami pooled in a single preparation.

G  Expression of immediate early genes and neuropeptides as analyzed by qRT–PCR in DIV8 primary hypothalamic neurons treated with BDNF (100 ng/ml) for 1.5 h. The log2-fold change in expression relative to the expression level in vehicle-treated neurons is shown. n = 4 technical replicates/treatment; data are from two neuronal culture preparations; 8-9 hypothalami pooled in a single preparation.

Data information: Data in (B, D, F, and G) are shown as mean ± SEM and analyzed using two-way ANOVA with Bonferroni post-test (B) or a two-tailed unpaired *t*-test (D, F, G). *$P < 0.05$, **$P < 0.01$, ***$P < 0.001$, ****$P < 0.0001$.

argue for a cell-autonomous function of the two receptors in regulation of AgRP expression.

The chronic increase in AgRP expression was documented in S1/3 KO mice as early as at 8 weeks of age and was preserved in aged mice (10 months). By magnitude, this increase is comparable to the change in Agrp transcription (50%) observed in rodents upon prolonged fasting [45–47]. The increase in Agrp expression in S1/3 KO mice was independent not only of age, but also of feeding status. This observation strongly suggests that the effect of SORCS1/3 ablation on Agrp transcription is not simply a consequence of developing leptin resistance, since leptin levels are greatly affected by fasting and age [48]. Rather, the attenuated capacity of leptin to decrease food intake in S1/3 KO mice, as seen at 15 weeks of age (Fig 6A), can be explained by the chronically increased AgRP levels in these animals. Importantly, enhanced expression of AgRP in S1/3 KO mice is already seen at the onset of adulthood (8 weeks), which likely precedes any age- and adiposity-related decline in leptin sensitivity [49].

Interestingly, the phenotype of S1/3 KO mice bears close resemblance to that of MC3R-deficient animals [50,51]. AgRP acts as an inverse agonist of melanocortin receptors MC3R and MC4R, blocking their activity on the second-order neurons throughout the CNS [22,52]. While lack of MC4R leads to overt obesity with a simultaneous rise in fat and lean tissue mass [53], deficiency in MC3R (as well as in SORCS1/3) results in an increased fat and a concomitantly decreased lean mass, and in aberrant nutrient partitioning in the absence of classical obesity [50,51]. These similarities place SORCS1 and SORCS3 as modulators of the melanocortin system that act in a subset of AgRP/NPY neurons projecting to brain areas with the predominant MC3R expression.

We identified an increased responsiveness to BDNF as a plausible cause of aberrant neuropeptide expression in S1/3 KO neurons. Among its many functions, BDNF emerges as an important regulator of energy expenditure [27,28]. With relevance to this study, BDNF modulates expression and secretion of various neuropeptides (e.g., oxytocin, NPY, somatostatin) in cultured hypothalamic and cortical neurons as well as in vivo [34–37,54]. Now, we show robust upregulation of Agrp transcription in primary hypothalamic neurons treated with BDNF, documenting AgRP as a novel target of BDNF signaling (Fig 8F). In agreement with our finding, virus-mediated overexpression of BDNF in the hypothalamus leads to a 15-fold increase in Agrp mRNA levels in mice [55]. We also show expression of the BDNF receptor TrkB in FACS-sorted NPY/AgRP neurons (Fig 8E), arguing that the effect of BDNF on this neuronal population is, at least in part, cell autonomous. In support of our conclusion, TrkB transcripts were detected at relatively high levels in NPY/AgRP neurons by high-throughput transcriptomics and found to increase by 40% with fasting [33].

Primary cortical neurons and hypothalamic slices from S1/3 KO mice show increased sensitivity to exogenously applied BDNF, as evidenced by the enhanced TrkB phosphorylation (Fig 8A–D). Increased responsiveness to BDNF may be explained by the altered subcellular localization of TrkB in primary neurons and in brains lacking SORCS1 and SORCS3 (Fig 7). Given that both SORCS1 and SORCS3 bind TrkB, we proposed a role for the two receptors in intracellular trafficking of TrkB. Our model is in agreement with the function of other VPS10P domain receptors that control neuronal sorting of various neurotrophin receptors [29–31]. In contrast to other VPS10P domain receptors, that typically promote

neurotrophin signals, SORCS1 and SORCS3 appear distinct as they attenuate BDNF signaling, presumably by reducing the active pool of TrkB on the surface of neuronal processes. Our data do not exclude that aberrant BDNF signaling in hypothalamic cell populations other than NPY/AgRP neurons, or in higher brain structures, may contribute to the phenotype of S1/3 KO mice. However, considering the prominent hypothalamic expression pattern of SORCS1 and SORCS3, and the chronic increase in AgRP levels in S1/3 KO mice, loss of the two sorting receptors from NPY/AgRP neurons appears to be a primary contributor to the increased feeding behavior and worsened nutrient partitioning in these animals.

How does BDNF/TrkB signaling affect Agrp transcription? A plausible mediator of this action is KLF4. Its expression is rapidly induced by BDNF in established cell lines and primary hippocampal neurons [38–40], as well as in primary hypothalamic neurons in this study (Fig 8G). KLF4 is produced in NPY/AgRP neurons and directly activates Agrp transcription [17,18]. Increased levels of KLF4 in the Arc of S1/3 KO mice coincide with upregulated AgRP expression and can be explained by enhanced hypothalamic BDNF signaling in the absence of SORCS1/3. Despite having documented a genotype-specific difference in vivo, we failed to recapitulate robust differences in the induction of Klf4 and Agrp transcription by BDNF comparing primary hypothalamic S1/3 KO and WT neurons (data not shown). In vivo, BDNF is released locally and in limited amounts, making axonal/dendritic localization of TrkB crucial for control of signal perception. Possibly, stimulation of cultured postnatal neurons with an excess of BDNF obscures the modulatory effects of SORCS1 and SORCS3 activities on TrkB trafficking and signal transmission in vitro.

Given the stimulatory role of BDNF on orexigenic neuropeptide expression (Fig 8F), the well-established anorexigenic effects of acute BDNF delivery in rodents seem contradictory. For example, hypothalamic overexpression of BDNF or acute delivery to PVH or VMH causes a strong decrease in food intake and loss of body weight, whereas chronic reduction of TrkB or BDNF levels in TrkB- and BDNF-heterozygous mice increases body weight and adiposity [56–58]. Conceivably, BDNF action in specific neuronal populations may have different, if not opposing effects on energy metabolism. Supporting this notion, the modulation of the BDNF pathway by TrkB agonists in vivo resulted in reduced, unchanged, or even increased food intake in rodents, depending on the pharmacological agent and the route of its administration [59,60]. Also, peripheral administration of BDNF in non-human primates led to a gain in appetite and increased adiposity [61]. In the end, seemingly contradictory systemic effects of manipulating the BDNF/TrkB pathway may largely depend on the distinct experimental conditions eliciting different effects in specific neuronal populations.

In conclusion, our findings identified a novel role for VPS10P domain receptors SORCS1 and SORCS3 in hypothalamic control of metabolism through regulation of orexigenic peptide production. Clearly, different scenarios may be envisioned whereby these multifunctional receptors impact energy homeostasis. Our data suggest a model in which SORCS1 and SORCS3 functionally interact with TrkB to alter its subcellular localization and, thereby, reduce responsiveness of this receptor to BDNF signals in hypothalamic neurons. Obviously, further studies will be required to ultimately resolve the molecular mechanism(s) underlying the action of these sorting receptors in regulation of energy balance.

# Materials and Methods

### Mouse models

Mice with targeted disruption of *Sorcs3* (S3 KO) have been described before [14]. To produce mice with combined deficiencies for *Sorcs3* and *Sorcs1* (S1/3 KO), murine embryonic stem (ES) cells heterozygous for the floxed *Sorcs3* allele (*Sorcs3*$^{lox/+}$) were transfected with a targeting construct in which exon 1 of *Sorcs1* was replaced by a puromycin expression cassette flanked by FRT sites (Appendix Fig S1A). Positive ES cell clones were injected into blastocysts to generate germ-line chimera carrying the targeted *Sorcs1* and *Sorcs3* alleles on the same chromosome 19. Subsequently, the mice were crossed with the Cre deleter strain (B6.C-Tg$^{CMV-cre}$1Cgn/J; Jackson Laboratories) to remove the floxed exon 1 from the targeted *Sorcs3* locus and with Flp deleter animals (B6.129S4-*Gt (ROSA)26Sor*$^{tm1(FLP1)Dym}$/RainJ; Jackson Laboratories) to remove the puromycin cassette in *Sorcs1*. The resulting (*Sorcs1*$^{+/-}$, *Sorcs3*$^{+/-}$) mice were backcrossed to C57Bl/6N wild-type animals for seven generations. To generate the single *Sorcs1*-targeted line (S1 KO), ES cell clones carrying the targeted *Sorcs1* allele on the *Sorcs3*$^{+/+}$ background were selected from the same ES cell targeting experiment and used to generate germ-line chimeras. All single or doubly targeted mouse strains were kept by breeding of heterozygous animals on an inbred C57Bl/6N genetic background (Appendix Fig S1B).

S1/3 KO mice expressing humanized renilla (hr)GFP in AgRP/NPY neurons were generated by crossing *Sorcs1/3*$^{-/-}$ and *Npy*-GFP mice (6.FVB-Tg(Npy-hrGFP)1Lowl/J; stock number 006417; The Jackson Laboratory). The resulting *Sorcs1/3*$^{+/-}$ mice carrying one copy of the NPY-GFP transgene (*Npy*-GFP$^{Tg/-}$) were bred with (*Sorcs1/3*$^{+/-}$) animals to derive (*Sorcs1/3*$^{-/-}$; *Npy*-GFP$^{Tg/-}$) (Npy/S1/3 KO) and littermate controls (*Sorcs1/3*$^{+/+}$; *Npy*-GFP$^{Tg/-}$) (Npy/WT) for analysis.

Mice were kept under stable environmental conditions on a 12/12-h light/dark cycle and fed a standard chow diet providing 66 kcal% from carbohydrates, 23 kcal% from protein, and 11 kcal% from fat (Sniff Spezialdiäten GmbH). Analyses were performed in female mice comparing mutant animals with their matched littermate controls. Age of the mice used in various experiments is specified in the respective figure legends. All animal experimentation was performed in accordance with institutional guidelines following approval by the local authorities of the State of Berlin (X9012/12, G0339/12).

### Metabolic phenotyping of mice

Body composition was assessed in conscious mice by nuclear magnetic resonance imaging. Food intake, gas exchange, and spontaneous locomotor activity were recorded in metabolic cages (TSE PhenoMaster System, TSE Systems). In detail, animals were kept in metabolic cages individually for four consecutive days, with the first day being considered as adaptation period (not analyzed). Parameters for each mouse were recorded at 8-min intervals. Data were analyzed according to the guidelines provided in [62]. Energy expenditure was determined by indirect gas calorimetry and adjusted for lean body mass by analysis of covariance (ANCOVA). Respiratory exchange ratio was calculated as the ratio between volumes of $CO_2$ produced and $O_2$ consumed. Spontaneous locomotor activity was measured by recording interruptions of infrared light beams emitted along the x- and y-axis of each cage. Analysis of glucose metabolism of mice is detailed in Appendix Supplementary Methods.

### Measurement of hormones

For measurement of blood hormones, mice were fasted overnight and blood was collected from the facial vein of non-anesthetized animals (for insulin and corticosterone), or by cardiac puncture from terminally anesthetized animals (for adrenocorticotropic hormone (ACTH) and growth hormone, leptin) in EDTA-treated tubes. Urine samples were collected overnight in metabolic cages. All hormones, except for ACTH, were measured using commercially available ELISA (Crystal Chem Inc. #90080; Enzo Life Sciences, Inc. ADI-900-097; Cloud-Clone Corp. SEA044Mu; Crystal Chem #90030). Plasma ACTH and urinary catecholamines were measured by diagnostic laboratory Biocontrol.

### Leptin sensitivity test

To measure effect of leptin on food intake, 14–16-week-old mice were i. p. injected with vehicle (PBS) on the first day of experiment (9 am and 6 pm) and with recombinant mouse leptin (2.5 mg/kg body weight; Sigma L3772) on the second day (9 am and 6 pm). Food intake was measured over a 24-h period after the first vehicle or leptin injection.

### Lipolytic activity in adipose tissue explants

Lipolytic activity of perigonadal adipose tissue was assessed *in vitro* essentially as described [63]. In brief, perigonadal adipose tissue pads were dissected from overnight fasted 22-week-old mice, cut into pieces (~70 mg), washed in PBS, and incubated for 1 h at 37°C in DMEM with 2% fatty acid-free bovine serum albumin. Thereafter, tissue explants were moved to the fresh medium with or without 10 μM isoproterenol for 1 h. Medium from the explants was analyzed for glycerol using a commercially available kit (Biovision #K630-100). Glycerol released in the medium was normalized to tissue protein content. Protein extraction from tissue explants was performed as described [64].

### Quantitative RT–PCR

Total RNA was extracted from tissue and cell lysates using TRIzol reagent and purified with RNeasy Mini/Micro Kit (Qiagen). Reversely transcribed cDNA from total RNA was subjected to qRT–PCR using the following Taqman Gene Expression Assays: *Sorcs1* ex 23 and 24 (Mm00491259_m1), *Sorcs3* ex 5 and 6 (Mm00458702_m1), *Agrp* (Mm00475829_g1), *Npy* (Mm01410146_m1), *Pomc* (Mm00435874_m1), *Crh* (Mm01293920_s1), *Trh* (Mm01182425_g1), *Klf4* (Mm00516104_m1), *Ucp1* (Mm01244861_m1), *Cidea* (Mm00432554_m1), *Ppargc1a* (Mm01208835_m1), *Adrb3* (Mm02601819_g1), *Pnpla2* (Mm00503040_m1), *Lipe* (Mm00495359_m1), *Sst* (Mm00436671_m1), *Ghrh* (Mm00439100_m1), *c-fos* (Mm00487425_m1), *Actb* (Mm02619580_g1), *Gapdh* (Mm99999915_g1). Fold change in gene expression was calculated using the cycle threshold (CT)

comparative method ($2^{-ddCT}$) normalizing to *Gapdh* or *Actb* CT values [65].

## Histology and immunodetection

Perigonadal adipose tissue pads and quadriceps muscle were dissected from 14- to 15-week-old mice, fixed in 4% paraformaldehyde (PFA), and embedded in paraffin. Tissue sections were stained with hematoxylin and eosin (H&E) and imaged using a Leica TSC SP2 microscope. For quantification of adipocyte size distribution, a minimum of three sections per mouse across adipose tissue depot (332–551 adipocytes) were analyzed. Adipocyte pixel area was measured using Fiji software (ImageJ) and converted to adipocyte diameter.

For immunodetection, mouse brains were dissected from 6- to 10-week-old intracardially perfused mice. After post-fixation (24 h) and cryopreservation in 30% sucrose/PBS, brains were cut in 30-μm-thick coronal sections using a sliding microtome. Free-floating sections across the arcuate nucleus were stained for KLF4 (Cat No 4038, Cell Signaling; dilution 1:500). Primary antibodies were visualized using Alexa Fluor 555 secondary antibody conjugates. AgRP/NPY neurons were identified by native hrGFP fluorescence from the *Npy*-GFP^{Tg/−} transgene. Images were acquired using confocal laser scanning microscope (Zeiss NLO 710). KLF4-positive areas were quantified as percentage of GFP-positive area using Fiji software (ImageJ). For quantification of KLF-positive area, two sections per mouse in the middle part of the arcuate nucleus (the region 1.82–1.94 mm caudal to bregma) were analyzed. The total number of AgRP/NPY neurons in the arcuate nucleus was obtained by manual count of all GFP-positive cell bodies in every forth section of the nucleus and subsequent multiplication times four.

The following additional antibodies were used in this study for immunohistology and Western blot analyses: SORCS1 (NoNBP1-86096, Novus Biologicals; dilution 1:200), SORCS3 (MAB3067, R&D Systems; dilution 1:200), KLF4 (No 4038, Cell Signaling; dilution 1:500), pTrkA/pTrkB (Y706/707) (No 4621, Cell Signaling; dilution 1:1,000), TrkB (No ab18987, Abcam; dilution 1:1,000), tubulin (No CP06, EMD Millipore; 1:5,000), GAPDH (No ab9484, Abcam; dilution 1:1,000), PSD95 (No 3409, Cell Signaling; dilution 1:1,000), synaptophysin (No 101011, Synaptic Systems; dilution 1:5,000), Golgin 97 (No 13192, Cell Signaling; dilution 1:1,000).

## In situ hybridization

Adult female C57BL/6J mice were sacrificed by cervical luxation, and brains flash-frozen using liquid nitrogen and stored at −80°C until cryosectioning. *In vitro* transcription and RNA labeling with α-^{35}S-UTP was performed according to the manufacturer's instructions (Promega, Madison, WI). *In situ* hybridizations were performed as described before [12]. In brief, 20-μm parallel coronal cryosections were fixed, acetylated, dehydrated, subjected to *in situ* hybridization at 55°C for 18 h, followed by RNAseA treatment and a high stringency wash in 0.1× saline sodium citrate buffer at 55°C. After dehydration slides were exposed to X-ray films for 1–14 days. All ISH probes were cloned into pBSK (Stratagene, La Jolla, CA) and comprised the following nucleotides: SorCS1, 1023–1545 (NM_021377); SorCS3, 1850–2521 (AF276314); NPY, 1–491 (BC043012); and AgRP, 86–586 (NM_007427.3).

## Brain membrane fractionation

Brain subcellular fractionations were performed according to a protocol adapted from [66,67]. In brief, brains from 7- to 12-week-old mice were homogenized in HEPES-buffered sucrose (0.32 M sucrose, 4 mM HEPES, protease, and phosphatase inhibitors, pH 7.4). Nuclear fractions were removed by centrifugation (1,000× *g* 10 min) and post-nuclear supernatant (S1) further centrifuged (10,000× *g* 15 min) to pellet the crude synaptosomal fraction (P2). The resulting supernatant was subjected to ultracentrifugation (150,000× *g* 30 min) to yield the light membrane-enriched fraction (P3). The P2 fraction was washed in HEPES-buffered saline, subjected to hypo-osmotic lysis in 4 mM HEPES and subsequent centrifugation (25,000× *g* 20 min) to pellet the synaptosomal membrane fraction (LP1). The corresponding supernatant was centrifuged (165,000× *g* 2 h) to obtain synaptic vesicles fraction (LP2). LP1 pellet was resuspended in HEPES-buffered saline and layered on top of discontinuous sucrose gradient. The gradient was centrifuged (150,000× *g* 2 h) to recover synaptic plasma membranes (SPM) at the interphase between 1.0 and 1.2 M sucrose. The SPM fraction was further solubilized in 50 mM HEPES containing 0.5% Triton X-100. Post-synaptic density membranes (PSD) were pelleted by centrifugation (200,000× *g* 20 min) from the solubilized SPM.

## Primary cortical and hypothalamic neurons

Primary cortical and hypothalamic neurons were prepared from newborn S1/3 KO and WT mice using enzymatic digestion with papain. Neurons were plated on poly-D-lysine-coated plates ($0.8 \times 10^6$ live cells/cm$^2$ for cortical neurons, $0.4 \times 10^6$ live cells/cm$^2$ for hypothalamic neurons). Neurons were maintained in Neurobasal medium (Invitrogen) supplemented with B27, GlutaMAX, and penicillin/streptomycin (Invitrogen). Medium for hypothalamic neurons was additionally supplemented with 5% heat-inactivated horse serum and cytosine β-d-arabinofuranoside (AraC, added on DIV3 at 1 μM). The medium for cortical neurons was not replaced during the entire culture period but the medium for hypothalamic neurons was renewed every 3 days by replacing 50% with fresh medium. Where stated, neurons were stimulated with recombinant BDNF (R&D Systems) added to the conditioned medium at a final concentration 100 ng/ml. Neurons were lysed at the indicated time points for either protein analysis or RNA extraction.

## Surface proteome analysis

Mixed hippocampal/cortical neuronal cultures were prepared from newborn WT or S1/3 KO mice and plated on poly-D-lysine-coated plates ($5 \times 10^6$ cells/10-cm plate). Cell surface biotinylation was performed at DIV10-12 using EZ-Link™ Sulfo-NHS-SS-Biotin (Thermo Fisher Scientific) according to published protocols [68]. Biotinylated proteins were pulled down with Neutravidin slurry (Pierce). After washing, the beads were snap-frozen and stored at −80°C until mass spectrometry analysis. For mass spectrometry, each sample was run on a stacking SDS–PAGE collecting all proteins in a single band. After Coomassie staining, the gel pieces were minced and digested with trypsin in an automated fashion using a PAL robot (Axel Semrau/CTC Analytics) [69]. Peptides were extracted with extraction buffer (80% acetonitrile, 0.1% [v/v]

formic acid) and dried in a speed-vac followed by purification on C18 stage-tips [70]. The eluted peptides were dried in a speed-vac and resuspended in 3% acetonitrile, 0.1% (v/v) formic acid for LC-MS measurement.

The samples were measured by LC-MS/MS on a Q Exactive Plus mass spectrometer (Thermo) connected to a Proxeon nano-LC system (Thermo). The peptides were separated on an in-house prepared nano-LC column (0.074 mm × 250 mm, 3 μm Reprosil C18, Dr Maisch GmbH) using a flow rate of 0.25 μl/min. MS acquisition was performed at a resolution of 70,000 in the scan range from 300 to 1,700 m/z. MS2 scans were carried out at a resolution of 15,500 with the isolation window of 2.0 m/z. Dynamic exclusion was set to 30 s, and the normalized collision energy was specified to 26 eV.

For analysis, the MaxQuant software package version 1.5.2.8 [71] was used. A FDR of 0.01 was applied for peptides and proteins, and the Andromeda search was performed using a *Mus musculus* Uniprot database (August 2014). MS intensities were normalized by the MaxLFQ algorithm implemented in MaxQuant [72]. MaxLFQ-normalized intensities among the replicates of the groups to be related were used for statistical comparison. Proteins were considered as specifically enriched for a group if they fulfilled a defined fold change (FC) of the averaged normalized intensities (see result section) and a *P*-value from a Student's *t*-test < 0.05 for comparison of two groups. For data visualization, the –log10 (*P*-value) was plotted against the log2 (FC) using *R* (www.r-project.org).

### Acute hypothalamic slices

Brains were dissected from S1/3 KO and WT mice at post-natal day 5 (P5). Tissue blocks containing hypothalami were obtained using the McIlwain tissue chopper set at 250 μm. Tissue slices were equilibrated in carbonated artificial cerebral spinal fluid (ACSF) at 37°C for 45 min, with the incubation medium changed every 15 min. Thereafter, the slices were stimulated for 1 h with recombinant BDNF at a final concentration of 200 ng/ml ACSF. Subsequently, the tissues were lysed in RIPA buffer and analyzed by Western blotting.

### Co-immunoprecipitation experiments

Chinese hamster ovary cells were transfected with expression constructs encoding rat TrkB and murine SORCS1 or SORCS3 using Lipofectamine™ 2000 reagent (Invitrogen). Cells were lysed 48 h post-transfection in 0.3% CHAPS containing IP-buffer (20 mM HEPES, 150 mM NaCl). Immunoprecipitation was performed using anti-TrkB antibody (ab18987, Abcam) for 30 min at 4°C. Antibody/protein complexes were isolated from the lysates using immobilized protein G agarose beads and subjected to SDS–PAGE and Western blotting according to standard protocols.

### Fluorescence-activated cell sorting (FACS) of neurons from mouse brain

*Npy*-GFP mice were sacrificed after an overnight fast, and the brains were rapidly extracted. The area containing the arcuate nucleus was microdissected based on anatomical landmarks. Tissue from 10 mice was pooled. To obtain a single-cell suspension, tissue was subjected to enzymatic digestion according to the published protocol using papain dissociation system (Worthington) [73]. GFP$^+$ cells were sorted on a BD FACSAria™ IIu directly in TRIzol reagent for subsequent mRNA extraction. The expression of target genes in GFP$^+$ neurons was related to their general expression in the arcuate nucleus (i.e., in the dissociated cell suspension taken before FACS sorting).

### Statistical analyses

For all *in vivo* experiments, an indicated *n*-number is the number of mice per group used in an experiment. Each mouse represents a statistically independent experimental unit, which is treated accordingly as an independent value in the statistical analyses. Statistical analyses were performed using GraphPad Prism software. For comparison between the two experimental groups, a two-tailed unpaired *t*-test was used. For comparison between three or more groups, one- or two-way ANOVA with Bonferroni post-test was applied. Energy expenditure was analyzed by ANCOVA using the online tool provided by NIDDK Mouse Metabolic Phenotyping Centers (supported by grant DK076169). The statistical tests used for the experiments are specified in the respective figure legends.

**Expanded View** for this article is available online.

### Acknowledgements
We are indebted to K. Kampf, A. Eisenmann, M. Schmeisser, and T. Pasternack for expert technical assistance. Also, we are grateful to Hans-Peter Rahn (MDC) and Anje Sporbert (MDC) for their valuable help in performing the FACS and live-cell imaging experiments. Studies were funded in part by the European Research Council (BeyOND No. 335692), the Helmholtz Association (iCEMED), and the Berlin Institute of Health (Collaborative Research Group 11220008).

### Author contributions
AS, ARM, GH, OP, TR, AS, PSB, TB designed and performed experiments. AS, GD, MNP, DS, and TEW designed experiments and evaluated data. AS and TEW wrote the manuscript.

### Conflict of interest
The authors declare that they have no conflict of interest.

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
