## [Review Process File · EMBO Reports]

SORCS1 and SORCS3 control energy balance and orexigenic peptide production

Aygul Subkhangulova, Anna R Malik, Guido Hermey, Oliver Popp, Gunnar Dittmar, Thomas Rathjen, Matthew N Poy, Alexander Stumpf, Prateep Sanker Beed, Dietmar Schmitz, Tilman Breiderhoff, Thomas E Willnow

Review timeline:

Submission date:	12 July 2017
Editorial Decision:	25 August 2017
Resubmission:	10 December 2017
Editorial Decision:	9 January 2018
Revision received:	15 January 2018
Accepted:	22 January 2018

Editor: Esther Schnapp

Transaction Report:

1st Editorial Decision

25 August 2017

Thank you for your patience while your manuscript was peer-reviewed at EMBO reports. I apologize for the unusual delay in the decision process due to the current holiday season. We have now received the full set of referee reports on your study that is copied below, in addition to cross-comments and input from an external advisor, whom I contacted because of the divergent referee opinions.

Taken all together, I am sorry to say that we cannot offer to publish your manuscript at this stage. As you will see, while the referees and the advisor acknowledge that the S1/S3 KO mouse phenotype is interesting, they also point out that the underlying mechanism is insufficiently worked out. Referees 1 and 2 indicate that a link between TrkB and AgRP signaling would need to be strengthened, and that a role for AgRP in the S1/S3 KO phenotype would need to be demonstrated. While referee 3 is more positive, the advisor consulted agrees that the underlying mechanism has not been conclusively demonstrated. I therefore return the manuscript to you at this point with the note that we cannot offer to publish it.

However, in case you feel that you can address the referee concerns in a timely manner and obtain data that would considerably strengthen the message of the study, i.e. the underlying mechanism, then we would have no objection to consider a new manuscript on the same topic in the near future. Please note that if you were to send a new manuscript this would be treated as a new submission rather than a revision and would be reviewed afresh, also with respect to the literature and the novelty of your findings at the time of resubmission.

At this stage of analysis, I am sorry to have to disappoint you. I nevertheless hope, that the referee comments will be helpful in your continued work in this area, and I thank you once more for your interest in our journal.

REFeree REPORTS

Referee #1:

This manuscript characterizes the phenotypes of mice with a dual knockout of the *Sorcs1* and *Sorcs3* genes. The data are of high quality, but a concern is that the phenotypes are quite modest and raise the question of whether or not the effect sizes are physiologically significant. The manuscript concludes with evidence that these genes interact with *TrkB* and the authors propose that this might desensitize neurons for BDNF signaling.

1. An example of a modest phenotype is the Rq measurements shown in Figure 2A & B. The Rq changes from about 0.92 to about 0.96. This is a very small change and the narration of the manuscript treats it as a real trait in the mice. In addition, these are high values for a mouse not on a very high carbohydrate diet. I could not find information about the diet.
2. The expression change of *Agrp* in Fig. 4 is approximately 1.6-fold. Again, the authors have not placed this into a physiological context. Is this a physiologically significant difference?
3. The interaction of *Sorcs1* with *TrkB* in Figure 6 is convincing. Based on these results, the authors propose that the interaction of *TrkB* with *Sorcs1* results in desensitization of *Agrp* neurons and increased induction of *KLF4* via BDNF. It would be good if the authors could demonstrate in any model system, if possible, that this pathway can occur.

Referee #2:

In this manuscript the authors investigated the role of sorting receptors *SORCS1* and *SORCS3* in metabolism. They found that knocking out either the *Sorcs1* or *Sorcs3* gene increased adiposity while decreasing body weight and lean mass. The role of these two genes in body composition is additive, because *Sorcs1/Sorcs3* double knockouts (S1/S3 KO) displayed more robust phenotypes than single-gene knockouts. Their data further indicate that increased adiposity is due to increased food intake. These interesting results demonstrate an important role of the two sorting receptors in nutrition partitioning and energy homeostasis. The authors then focused on elucidating the mechanism underlying increased food intake. They proposed that *SORCS1* and *SORCS3* normally impair *TrkB* surface expression in *AgRP* neurons to reduce food intake. This claim is not consistent with literature and also is not supported by the presented data. Overall, the manuscript presents an interesting metabolic phenotype in S1/S3 KO mice; however, mechanistic studies are preliminary and interpretation of the data from these studies are problematic.

Specific comments:

1. Reduction in body weight of S1/S3 KO mice appears to occur during the first three postnatal weeks (Fig. EV2A), indicating that the two sorting receptors are important for postnatal development of muscles and/or bones. No experiments were carried out to address this issue.
2. Both *Sorcs1* and *Sorcs3* are expressed in several hypothalamic nuclei. It is unclear if the two proteins are expressed in the same cells or distinct sets of cells in each of these nuclei. Also, no data are presented to show that the two receptors are expressed in neurons rather than glial cells.
3. *AgRP* expression was upregulated in both fed and fasted S1/S3 KO mice in comparison with WT mice (Fig. 4A and B). This is likely a response to leptin resistance, as the authors showed that leptin administration was less effective in reducing food intake in S1/S3 KO mice than in WT mice (Fig. 3A). To show that *SORCS1/SORCS3* are involved in the regulation of *AgRP* gene expression, the authors should use young S1/S3 KO mice that have not developed leptin resistance.
4. Even if *AgRP* expression is upregulated in young S1/S3 KO mice, it is unlikely that the upregulation is due to increased BDNF-*TrkB* signaling in *AgRP* neurons. It has been shown that only a small fraction of *AgRP* neurons expresses *TrkB* (Liao et al., 2015, *Mol Metab* 4: 471-482). The observation that BDNF overexpression increased levels of *AgRP* mRNA (Cao et al., 2009, *Nat Med* 15: 447-454) should be a compensatory response to energy deficits. To demonstrate that increased food intake and adiposity in S1/S3 KO mice results from *AgRP* upregulation, the authors should show that blocking *AgRP* upregulation can abolish the metabolic phenotype in S1/S3 KO mice.

5. The model "whereby SORCS1 and -3 bind TrkB to impair surface exposure of this receptor, desensitizing AgRP neurons for BDNF signals" is against a large body of literature. Increasing BDNF-TrkB signaling has been consistently shown to reduce food intake and adiposity in mice.
6. Quantification is needed for surface TrkB in Fig. 6.
7. It is important to indicate the age of mice used for each experiment. No age information is available for Fig. 4, Fig. 6, and Fig. EV2.

Referee #3:

This is an excellent scientific manuscript from Subkhangulova, Willnow, and colleagues. The manuscript is very well written, the analyses are clear and direct, and the metabolic and molecular phenotyping of the SORCS1 and 3 (and combined) mutant mice is elegant and comprehensive. The authors reveal that the SORCS1 and 3 sorting receptors synergize with the BDNF-TrkB and melanocortin pathways to drive excessive energy intake and adiposity in mice, without producing other typical metabolic dysregulation associate with positive energy balance in rodents (e.g., insulin resistance, increased body weight). The manuscripts reads as if it has already been through extensive peer review (possibly at another journal), and therefore my comments are only minor.

[1] The explanation in the Discussion on why there are glucose tolerance deficits without concomitant insulin tolerance deficits is weak. The authors should expend on this critical finding in more depth. Is there a precedence for this? Is there an alternative explanation than 'decreased mass of muscle'?

[2] The authors should include in the Discussion the possibility that deleted SORCS signaling in the hippocampus may be contributing to the metabolic and energy intake phenotype. The SORCS3 expression in the hippocampus appears robust in Figure 3, and a number of recent high profile papers show that the hippocampus regulates feeding behavior (e.g., PMID 28461695 , 26666960). Moreover, TrkB is robustly expressed in hippocampal neurons. The possible role of the hippocampus should be briefly discussed.

[3] The authors very elegantly link the phenotypes with altered signaling in melanocortin central circuits (e.g., agrp neurons). In the Discussion they bring up a very interesting observation that the SORCS1-3 phenotype is much more similar to the MC3R-deficient mice compared to MC4R-deficient mice. I think this should be expanded on, and highlighted in more prominent parts of the manuscript than the 4th paragraph of the Discussion.

[4] The reduced body weight in the KO mice is a very important result that is virtually pushed under the rug. This should be highlighted/mentioned in the Abstract. Upon quick read of the abstract, one assumes an obese phenotype based on "chronic energy excess... enhanced food intake, decreased locomotor activity, ... increased adiposity, etc.", yet there is not mention of body weight. The reduced body weight phenotype must be highlighted and owned in the Abstract; otherwise I fear that the manuscript may get mis-cited in the future.

Cross-comments from referee 1:

The other reviews were excellent and raised points related to neuronal signaling that are important. I'm not an expert in neural signaling, so I defer to their judgement.

Cross-comments from referee 3:

I personally find the requests of Rev. #2 excessive. In each of the 1st 4 comments they are asking for substantial additional experiments (and 4 separate experiments). This is a paper - not a 5 year grant. Moreover, I don't find any of these 4 additional requested experiments - should they be completed - as filling any fatal flaws in the overall analysis, but rather, just filling that reviewer's extensive curiosity.

I agree with their comment #5 that the results are somewhat against the grain of the existing

literature with regards to BDNF-TrkB signaling and energy balance, but in my opinion this should in no way preclude publication. The authors do address this issue in the Discussion, but perhaps the authors should address this in more depth. However, to preclude publication primarily based on results that seem inconsistent with existing literature is a ridiculous notion that is fundamentally against scientific advancement. Many biological systems have opposing effects on energy balance depending on the neural locus, upstream or downstream pathways, model organism, etc.

If any additional experiments could be added, Rev. #1 (comment #3) and Rev. #2 (comment #4) touch on the same issue (directly linking sorcs1-trkb signaling with agrp signaling) that an additional experiment could potentially address and expand upon. However, Rev. #1 is correct in adding the phrase "if possible" as this would be extremely difficult to establish. Perhaps the authors can discuss the absence of direct demonstration of this pathway and offer more caution in their interpretation.

Advisor's comments:

I read the paper and the comments. The effects of these genetic manipulations on metabolic parameters are clear. I do not believe, however, that there is mechanistic explanation provided how these alterations emerge. Because SORCS1/3 are abundantly expressed in other brain regions (cortex, hippocampus), it cannot be excluded that the metabolic phenotype reported is the outcome of multiple effects originating outside of the hypothalamus. That is what I find to be the weakest point of the current paper.

1st Revision - authors' response

10 December 2017

COMMENTS TO REFEREE #1

General comments:

We very much appreciate this referee's comments concerning the high quality of our data. A major issue raised in this review concerns the physiological significance of the metabolic changes seen in our mouse models. Importantly, genetic perturbations of many pathways, playing a crucial role in regulation of energy and glucose metabolism, often lead to modest phenotypes in mice, unless additional environmental or genetic stressors are applied. Examples with relevance to this study include the obesity-associated melanocortin receptor 3 (*Mc3r*) (Butler et al. 2000, *Endocrinology* 141:3518-3521; Chen et al. 2000, *Nat Genet* 26:97-102). To address this reviewer's concern, we now analyzed a separate cohort of aged SORCS1/3 deficient mice and showed aggravation of parameters of the mutants' energy metabolism (including respiratory quotient) upon aging (Fig. 3). Thus, our original data combined with these new findings document that the joined loss of diabetes-associated SORCS1 and the related receptor SORCS3 results in an obvious metabolic phenotype, unequivocally documenting the (patho)physiological relevance of metabolic defects caused by SORCS1/3 deficiency.

Details of our new data are given in the point-by-point reply below.

Point 1:

"An example of a modest phenotype is the Rq measurements shown in Figure 2A & B. The Rq changes from about 0.92 to about 0.96. This is a very small change and the narration of the manuscript treats it as a real trait in the mice. In addition, these are high values for a mouse not on a very high carbohydrate diet. I could not find information about the diet."

In the original manuscript, we presented Rq measurements as mean value of day and night phases. This depiction was misleading as it obscured diurnal oscillation in RER. We apologize for this mistake.

We now split the values for RER between light and dark phases to enable proper comparison of our dataset to published results. Data in Fig. 2A/B show the following RER for 21-week-old WT mice: 0.86 ± 0.05 (day) and 0.97 ± 0.04 (night). These values fully agree with those reported by others (Chen et al., 2000, *Nat Genet* 26:97-102; Semjonous et al. 2009, *Int J Obes* 33:775-785; Joly-

Amado et al. 2012, EMBO J 31:4276-4288). At this age, the increase in RER as a consequence of SORCS1/3 ablation amounts to as much as half the size of the difference between the light and the dark cycle, which we consider a physiologically relevant effect size (Fig. 2A and B).

More importantly, we now report the metabolic phenotypes in SORCS1/3 mutant mice upon aging. Our investigations were guided by the notion that aging commonly aggravates metabolic dysfunctions. Gratifyingly, metabolic phenotypes described for mice at 21-weeks of age in the original manuscript (Fig. 2) were significantly worsened in mice at 9-10 months of age, even on a normal chow (Fig. 3).

As exemplified for RER, genotype-specific difference aggravated in 10-month old mice from 0.90 ± 0.03 (WT) to 0.99 ± 0.04 (S1/3 KO) during the day and from 0.99 ± 0.02 (WT) to 1.05 ± 0.02 (S1/3 KO) during the night. These data represent an average increase in RER in mutant mice from 4% at 21 weeks to 8% at 10 months as compared to age-matched littermates. For reference, obesity mouse models (such as mice lacking leptin, MC4R, or MC3R) show a similar change in RER only when shifted to a high fat diet (Butler et al. 2000, *Endocrinology* 141:3518-3521; Butler et al. 2001, *Nat Neurosci* 4:605-611). Also, mice lacking AgRP expression show a similar difference in RER as S1/3 mutants to their respective controls (Joly-Amado et al. 2012, EMBO J 31:4276-4288) (approximately 0.1).

Taken together, we strongly feel that our previous and our new data convincingly document the pathophysiological significance of SORCS1/3 deficiency for metabolic control, as these defects are comparable in magnitude to the experimental manipulation of established pathways in energy homeostasis, as through AgRP and leptin.

As a minor comment, we now include the description of the standard mouse chow used in this study in the method section (66 kcal% from carbohydrates, 23 kcal% from protein, and 11 kcal% from fat).

Point 2:

*“The expression change of *Agrp* in Fig. 4 is approximately 1.6-fold. Again, the authors have not placed this into a physiological context. Is this a physiologically significant difference?”*

By magnitude, the increase in *Agrp* expression by 1.6-fold (60%) is comparable to the change in *Agrp* transcription (50%) observed in rodents upon prolonged fasting (Korner et al. 2000, *Endocrinology* 141:2465-2471; Savontaus et al. 2002, *Brain Res* 958:130-138; Morrison et al. 2005, *Am J Physiol Endocrinol Metab* 289:E1051-1057). Since hunger is one of the strongest drivers of AgRP expression, the increase observed in S1/3 KO mice appears to be of physiological relevance.

Importantly, we now also report similar increases in *Agrp* levels in S1/3 KO mice as early as 8 weeks of age and also at an advanced age of 35 weeks (see Fig. 5B). These novel data establish pathophysiological increases in AgRP expression levels as a consistent feature of SORCS1/3 deficiency throughout the lifetime.

Point 3:

*“The interaction of *Sorcs1* with *TrkB* in Figure 6 is convincing. Based on these results, the authors propose that the interaction of *TrkB* with *Sorcs1* results in desensitization of *Agrp* neurons and increased induction of *KLF4* via *BDNF*. It would be good if the authors could demonstrate in any model system, if possible, that this pathway can occur.”*

We agree that establishing the molecular mechanisms whereby SORCS1 and -3 impact *Agrp* expression in Arc neurons is important. Based on our earlier data documenting the direct interaction of SORCS1 and -3 with TrkB by co-immunoprecipitation (revised Fig. 7D) and showing enhanced sensitivity of hypothalamic slices from S1/3 mutants to BDNF stimulation (revised Fig. 7C/D), we proposed SORCS1/3 as regulators of TrkB activity, ameliorating BDNF-dependent induction of *Agrp* through KLF4.

We now significantly extended our analyses of the molecular interactions between SORCS1/3 and TrkB by confirming altered subcellular localization of TrkB in the brain of mutant mice (Fig. 7D/E). We also obtained insights into the kinetics of TrkB trafficking in primary neurons lacking

SORCS1/3 using live cell imaging (Fig. 9A-D). Specifically, these novel findings suggest a shift of TrkB in mutant neurons from the retrograde to the anterograde moving vesicle pool. As such, this finding supports our model of SORCS1/3 acting as sorting factors for TrkB, reducing the active TrkB pool on the neuronal surface, thereby decreasing the sensitivity of hypothalamic neurons to BDNF signals (see Fig. 9E/F for a schematic).

As requested by reviewer 2 (point 2), we now also show co-expression of SORCS1 and -3 with TrkB in FACS-sorted AgRP neurons using qRT-PCR, substantiating that a molecular interaction between sorting receptors and TrkB can occur in this cell type (Fig. 8E).

Finally, we established primary hypothalamic neuron cultures to confirm the ability of BDNF to induce *Klf4* and *Agrp* (Fig. 8F/G). Despite extensive efforts, we failed to document consistent differences in BDNF-induced *Agrp* gene transcription comparing primary hypothalamic neurons from WT and S1/3 KO mice. Such a difference was easily documented in mice *in vivo* using qRT-PCR and quantitative immunohistology (Fig. 5). *In vivo*, BDNF is released locally and in limited amounts, making axonal/dendritic localization of TrkB crucial for control of BDNF signal reception. Possibly, stimulation of cultured postnatal neurons with an excess of BDNF obscures the modulatory effects of SORCS1 and -3 activity on TrkB trafficking and signal transmission *in vitro*. Obvious technical challenges in recapitulating intricate hypothalamic signaling pathways in cultured neurons have already been raised by referee 3 in a comment to this reviewer.

Taken together, we strongly feel that our extensive new data on the relevance of SORCS1/3 for TrkB sorting *in vivo* and *in vitro*, and on the BDNF-dependent induction of *Klf4* and *Agrp* in hypothalamic neurons provide substantial additional experimental support for our working model to explain the metabolic defects seen in SORCS1/3 deficient mice (Fig. 9E). Still, we phrased our conclusions in abstract and result sections now with more caution, and we revised the discussion section to indicate the caveats in fully recapitulating these complex molecular interactions in *in vitro* models (page 18, bottom). Also, we modified the title of the manuscript accordingly.

COMMENTS TO REFEREE #2:

General comments:

We appreciate the interest of this reviewer in our data and the acknowledgement of their potential importance. A major point raised by this referee concerns the apparent contradiction of our mouse phenotype (increased adiposity) with the ascribed actions of BDNF as anorexic and anti-adipogenic agent when overexpressed or exogenously applied to rodents. Another point concerns the question whether increases in AgRP levels in the SORCS1/3 deficient mouse model may represent a secondary consequence of leptin resistance rather than the primary cause of the metabolic disturbances. These comments are well taken and we now provide substantial new data, including phenotypic analysis of young (6-8 weeks) and of aged mice (35 weeks) that fully corroborate our initial findings. These novel data are detailed in the following.

Point 1:

“Reduction in body weight of S1/S3 KO mice appears to occur during the first three postnatal weeks (Fig. EV2A), indicating that the two sorting receptors are important for postnatal development of muscles and/or bones. No experiments were carried out to address this issue.”

As suggested, we now include a more detailed phenotypic description of mutant mice at a young age (6-8 weeks), including analysis of body composition by NMR and histology of muscle and white adipose tissue. These data fully confirm a change in body composition and an increase in adiposity in S1/3 KO mice as early as 6 weeks of age (Fig. EV2C), although, not surprisingly, the white adipose tissue was not yet hypertrophic at this young age (Fig. EV2D). Furthermore, we now confirm increased expression of AgRP in 8-weeks old mutant mice (Fig. 5B), similar to the increases reported for 21-weeks old animals in the original manuscript. Our findings are in line with our model whereby a chronic increase in hypothalamic expression of orexigenic peptides is a consistent feature of SORCS1/3 deficiency already seen at an early postnatal live.

The observed reduction in body weight can be explained by the decrease of lean mass, documented in S1/3 mice at different ages. It is not uncommon for genes involved in regulation of energy

metabolism, to also affect the lean mass and bone development. Examples of such genes with relevance to our study include melanocortin receptors 3 and 4, or the growth hormone receptor (Huszar et al. 1997, *Cell* 88:131-141; Butler et al. 2000, *Endocrinology* 141:3518-3521; Chen et al. 2000, *Nat Genet* 26:97-102; Berryman et al. 2004, *Growth Horm IGF Res* 14:309-318). Though the reduction of lean mass is an interesting aspect of the phenotype of S1/3 KO mice, the main focus of this study was to examine effects of receptor gene ablations on adiposity and glucose handling. Therefore, no further extensive experiments were carried out to investigate bone morphology.

Point 2:

“Both Sorcs1 and Sorcs3 are expressed in several hypothalamic nuclei. It is unclear if the two proteins are expressed in the same cells or distinct sets of cells in each of these nuclei. Also, no data are presented to show that the two receptors are expressed in neurons rather than glial cells.”

To clarify this important issue, we now performed q-RT-PCR analysis on FACS-sorted NPY/AgRP neurons and substantiated co-expression of SORCS1 and SORCS3 with TrkB in this neuronal cell population (Fig. 8E). In agreement with our original ISH data (Fig. 4E), SORCS3 transcripts are highly enriched in AgRP neurons as compared to other cells in the arcuate nucleus. Transcripts for SORCS1 and TrkB are also clearly detectable in AgRP neurons, albeit not enriched in this cell population. The latter finding is in line with the more wide-spread neuronal expression patterns of SORCS1 and TrkB. Taken together, our new data unequivocally document the co-expression of all three genes in AgRP neurons *in vivo*.

Our conclusions about the physiological relevance of SORCS1 and -3 expression in AgRP neurons are further supported by published data showing SORCS1 transcript levels in AgRP neurons to be highly dependent on the energy status and to increase approximately 10-fold by fasting. These findings clearly argue for a physiological role for SORCS1 in adaptation to hunger (Henry et al. 2015, *Elife* 4).

Point 3:

“AgRP expression was upregulated in both fed and fasted S1/S3 KO mice in comparison with WT mice (Fig. 4A and B). This is likely a response to leptin resistance, as the authors showed that leptin administration was less effective in reducing food intake in S1/S3 KO mice than in WT mice (Fig. 3A). To show that SORCS1/SORCS3 are involved in the regulation of AgRP gene expression, the authors should use young S1/S3 KO mice that have not developed leptin resistance.”

We now include new data that *Agrp* transcript levels are increased significantly in 8-weeks old S1/3 mutant mice (Fig. 5B). As stated by this referee and also reported in the literature, this early age likely precedes any age- and adiposity-related decline in leptin sensitivity (El-Haschimi et al. 2000, *J Clin Invest* 105:1827-1832).

As detailed in the revised discussion section (page 16, bottom), a chronic increase in AgRP expression is now documented in S1/3 KO mice as early as at 8 weeks of age and persists in aged mice (10 months) (Fig. 5B). By magnitude, this increase is comparable to the change in *Agrp* transcription (50%) observed in rodents upon prolonged fasting (Korner et al. 2000, *Endocrinology* 141:2465-2471; Savontaus et al. 2002, *Brain Res* 958:130-138; Morrison et al. 2005, *Am J Physiol Endocrinol Metab* 289:E1051-1057). The increase in *Agrp* expression in S1/3 KO mice is independent not only of age, but also of the feeding status. This observation argues that the effect of SORCS1/3 ablation on *Agrp* transcription is not merely a consequence of developing leptin resistance, since plasma leptin levels are greatly affected by fasting and age (Ahren et al. 1997, *Am J Physiol* 273:R113-120). Rather, the attenuated capacity of leptin to decrease food intake in S1/3 KO mice, as seen at 15 weeks of age (Fig. 6A), can be explained by the chronically increased AgRP levels in these animals.

Point 4:

*“Even if AgRP expression is upregulated in young S1/S3 KO mice, it is unlikely that the upregulation is due to increased BDNF-TrkB signaling in AgRP neurons. It has been shown that only a small fraction of AgRP neurons expresses TrkB (Liao et al., 2015, *Mol Metab* 4: 471-482). To demonstrate that increased food intake and adiposity in S1/S3 KO mice results from AgRP upregulation, the authors should show that blocking AgRP upregulation can abolish the metabolic phenotype in S1/S3 KO mice.”*

There are two issues raised in this comment.

The first one concerns the existence of a functional BDNF-TrkB pathway in AgRP neurons. As pointed out by this reviewer, only 8% of AgRP neurons was reported to express TrkB in adult mice (Liao et al. 2015, *Mol Metab* 4:471-482). It is important to note, however, that the number of TrkB+ neurons in this study was assessed using a reporter mouse strain. More precisely, TrkB+ cells were identified by the expression of dtTomato as a result of Cre-mediated excision of a STOP cassette. While we do not doubt the validity of this approach, we would like to point out that the absolute cell numbers obtained by this approach critically depend on the efficiency of Cre-mediated recombination. Potentially, the number of TrkB-expressing neurons in the adult brain may be underestimated considering the reduced efficiency of Cre recombination in adult tissues (Badea et al. 2009, *PLoS One* 4:e7859; Long and Rossi 2009, *PLoS One* 4:e5435).

Irrespective of the absolute number of TrkB+ cells in the Arc, we now clearly show expression of TrkB in FACS-sorted AgRP neurons (Fig. 8E). In this cell type, the receptor is co-expressed with SORCS1 and -3 (Fig. 8E). Our new findings are in agreement with published data from high-throughput transcriptomics, showing expression of TrkB in AgRP neurons at levels comparable to those of the leptin receptor, and documenting an approximate 40% increase in TrkB transcripts upon fasting (Henry et al. 2015, *Elife* 4). The latter finding argues for a functional relevance of TrkB in AgRP neurons in adaptation to hunger.

With respect to TrkB function in hypothalamic neurons, we now include new data that clearly document the ability of BDNF to induce AgRP expression in hypothalamic neurons. As shown in figure 8, treatment of primary hypothalamic neurons with BDNF results in a robust induction of *Agrp* transcription (Fig. 8F). *Agrp* transcription is preceded by a BDNF-induced increase in *Klf4* transcript levels (Fig. 8G), arguing for the ability of BDNF to induce AgRP expression through KLF4.

Taken together, we strongly feel that the ability of BDNF to induce AgRP expression in hypothalamic neurons and the hypersensitivity of this cell type to BDNF in the absence of SORCS1/3 provides substantial experimental support for our model that enhanced BDNF-induced transcription of *Agrp* is the primary cause of the chronic overexpression of this neuropeptide seen in S/3 KO mice throughout life.

The second issue raised in this comment questions whether AgRP is the primary contributor to the phenotype of S1/3 KO mice. To test this concept, a genetic model to reduce chronic AgRP overexpression and thereby rescue the SORCS1/3 phenotype is suggested.

We respectfully disagree with this reviewer that it may be technically feasible to rescue the metabolic phenotypes in SORCS1/3 mutants by dampening AgRP overexpression to physiological levels *in vivo*. Genetic ablation of AgRP leads to network-wide compensations in the mouse brain and to dysregulation of multiple hormonal axes, including the thyroid axis (Wortley et al. 2005, *Cell Metab* 2:421-427; Flier 2006, *Cell Metab* 3:83-85). Therefore, combining the complex consequences of the loss of AgRP with the ablation of SORCS1 and -3 does not seem an appropriate experimental solution to address this issue. Referee 3 made the same argument when commenting on the critique by this reviewer.

To this end, we have rephrased our discussion section to indicate that we do not exclude the contribution of other neuronal populations residing in hypothalamus and in higher brain structures to the onset and development of the described phenotypes (p. 18, top). Still, considering the prominent hypothalamic expression pattern of SORCS1 and -3, and the chronic increase in AgRP levels in S1/3 KO mice, loss of the two sorting receptors from NPY/AgRP neurons appears to be a primary cause of the increased feeding behavior and worsened nutrient partitioning in these animals.

Point 5.

“The model “whereby SORCS1 and -3 bind TrkB to impair surface exposure of this receptor, desensitizing AgRP neurons for BDNF signals” is against a large body of literature. Increasing BDNF-TrkB signaling has been consistently shown to reduce food intake and adiposity in mice.”

We agree that given the well-established anorexigenic effects of BDNF delivery in rodents, the stimulatory action of BDNF on AgRP expression (Fig. 8F) is surprising at first glance. However, to date, the pleiotropic effects of BDNF on energy intake and expenditure were studied mainly by means of global or hypothalamic overexpression of BDNF, its excessive exogenous delivery, or by global decrease of TrkB or BDNF levels in heterozygous mutant mice (reviewed in Rios, 2013, Trends Neurosci 36:83-90; Xu and Xie 2016, Nat Rev Neurosci 17:282-292). In contrast, our model displays a cell-autonomous defect in sensitivity to BDNF without impacting the global levels of the neurotrophin or its receptor. Conceivably, BDNF action in specific neuronal population(s) may have different, if not opposing, effects on energy metabolism. Supporting this statement, the modulation of the BDNF pathway *in vivo* by BDNF or TrkB agonists resulted in reduced, unchanged, or even increased food intake in rodents depending on the pharmacological agent and the route of its administration (Lin et al. 2008, PLoS One 3:e1900; Chan et al. 2015, Chem Biol 22:355-368). Treatment of non-human primates with BDNF elicited hyperphagia and weight gain (Lin et al. 2008, PLoS One 3:e1900). Complicating this issue even more, obesity status appears to affect the outcomes of BDNF administration on food intake. Thus, treatment of mice with low doses of BDNF led to the decrease in food intake only in HFD-fed mice (Nakagawa et al. 2003, Int J Obes Relat Metab Disord 27:557-565). Similarly, food intake was not changed by hypothalamic *Bdnf* gene transfer unless mice were made genetically obese by introduction of db/db mutation (Cao et al. 2009, Nat Med 15:447-454).

Taken together, while all experimental data clearly support an important role for the BDNF/TrkB pathway in central control of metabolism, the actions of this pathway may be more complex than anticipated. Thus, the physiological outcome of its manipulation may depend on the distinct experimental conditions used. While our data by no means refute work by others, our findings provide important new information concerning the complexity of this regulatory process that warrants publication. We now rephrased our discussion section to better put our findings in context with the literature (page 19).

Point 6.

“Quantification is needed for surface TrkB in Fig. 6.”

The quantification of the neuronal surface increase for TrkB had been provided by the mass spec analysis and determined as 1.57-fold increase (supplementary table 1). Details of the quantitative analysis in replicate experiments using the MaxQuant software are given in the supplementary method section.

In addition, we now present new quantitative data further substantiating altered trafficking and subcellular localization of TrkB in cultured neurons and in the brain *in vivo* using live cell imaging (Fig. 9A-D) and subcellular fractionation studies (Fig. 7D/E). Jointly, these quantitative data substantiate and refine our model whereby SORCS1 and -3 act as sorting factors for TrkB that retrogradely traffic BDNF along neurites to decrease surface exposure of this receptor and to desensitize AgRP neurons for BDNF signals (see Fig. 9E/F for our model).

Point 7.

“It is important to indicate the age of mice used for each experiment. No age information is available for Fig. 4, Fig. 6, and Fig. EV2.”

We apologize for this oversight. We now state the age of the mice in all figure legends.

COMMENTS TO REFEREE #3:

General comments:

My co-authors and I very much appreciate this referee’s enthusiasm for our study. We also appreciate his/her comments to points raised by the other referees, as well as the helpful suggestions how to rewrite abstract and discussion sections. These suggestions have been valuable in improving the clarity of our manuscript.

Point 1:

“The explanation in the Discussion on why there are glucose tolerance deficits without concomitant insulin tolerance deficits is weak. The authors should expend on this critical finding in more depth. Is there a precedence for this? Is there an alternative explanation than ‘decreased mass of muscle?’”

We agree that there are multiple potential explanations for the worsened glucose tolerance seen in S1/3 KO mice. To get a better insight in effects of the joint SORCS1/3 deficiency on glucose metabolism, we now expanded the phenotypical characterization of mutants upon aging (9-10 months). As these experiments revealed, old S1/3 KO mice develop impaired insulin secretion in response to a bolus of glucose (Fig. EV4C). This finding is of particular interest, because it is in agreement with data reported for mice deficient for SORCS1 only (Kebede et al. 2014, J Clin Invest 124:4240-4256). The fact that the cohort of old S1/3 KO mice was not hyperglycemic and showed no signs of glucose intolerance indicated that impairments in glucose-stimulated insulin secretion and mild worsening in insulin sensitivity were not sufficient to cause a diabetic phenotype in this mouse model. However, as we point out in the revised discussion section, the insulin secretion phenotype in S1/3 KO mice suggests involvement of SORCS3 (in addition to SORCS1) in regulation of pancreatic function. Since SORCS3 is not expressed in pancreatic islets, its impact on pancreatic hormones secretion may be indirect and mediated by its action in CNS. These considerations are now included in the revised discussion section (page 15).

Point 2:

“The authors should include in the Discussion the possibility that deleted SORCS signaling in the hippocampus may be contributing to the metabolic and energy intake phenotype.”

As suggested, we now discuss that the loss of SORCS1 and -3 action from hypothalamic cell populations other than NPY/AgRP neurons, as well as from higher brain structures, may contribute to the observed phenotypes (page 18 top).

Point 3:

“The authors very elegantly link the phenotypes with altered signaling in melanocortin central circuits. In the Discussion they bring up a very interesting observation that the SORCS1-3 phenotype is much more similar to the MC3R-deficient mice compared to MC4R-deficient mice. I think this should be expanded on, and highlighted in more prominent parts of the manuscript than the 4th paragraph of the Discussion.”

We now elaborate on the possible implications of SORCS1/3 action for MC3R activity and possible underlying molecular interactions (pages 16, bottom). Although quite speculative at present, we fully agree with this reviewer that the observed phenotypic resemblance of SORCS1/3 mutants with MC3R-deficient mice is intriguing and suggests so far unknown mechanisms that functionally distinguish various receptor pathways in the melanocortin central circuits.

Point 4:

“The reduced body weight in the KO mice is a very important result that is virtually pushed under the rug. This should be highlighted/mentioned in the Abstract.”

We now revised the abstract to also highlight reduced body weight as an important aspect of metabolic disturbances caused by combined deficiencies for SORCS1 and -3.

REPLY TO ADVISOR'S COMMENTS:

Point 1:

“I read the paper and the comments. The effects of these genetic manipulations on metabolic parameters are clear. I do not believe, however, that there is mechanistic explanation provided how these alterations emerge.”

Mice genetically deficient for SORCS1 and -3 suffer from a distinct metabolic phenotype characterized by elevated food intake, increased RER, decreased locomotor activity, reduced body temperature, and adipose tissue accumulation and hypertrophy. Similar metabolic disturbances are observed in mice with AgRP gain-of-function induced experimentally by injection or

overexpression of this neuropeptide. In the original manuscript, we documented increased levels of AgRP expression in the hypothalamus of S1/3 KO mice at 21 weeks of age (Fig. 5A). As requested by reviewer 2, we now confirmed chronically elevated levels of AgRP transcripts in young (8 weeks) and in aged mice (9-10 months) (Fig. 5B). The phenotypic similarities between SORCS1/3 deficient mice and AgRP gain-of-function models, and the consistent increase in AgRP levels seen in S1/3 mutants throughout life strongly argues for increased AgRP activity as the underlying cause of the metabolic defects in SORCS1/3 mutant animals.

Because AgRP is mainly expressed in NPY/AgRP neurons of the hypothalamus, we now confirmed expression of SORCS1 and -3 in FACS-sorted AgRP neurons, along with expression of its molecular target TrkB (Fig. 8E). We now also provide new data using subcellular fractionation and live cell imaging to confirm altered trafficking and subcellular localization of TrkB in neurons lacking SORCS1/3 *in vivo* (Fig. 7D/E) and *in vitro* (Fig. 9A-D). These data provide substantial additional evidence to support our model of SORCS1/3 as sorting factors for TrkB that reduce the active surface pool of TrkB (see Fig. 9E/F for a schematic model). Specifically, our data strongly suggest that the increased surface levels of TrkB in SORCS1/3-deficient neurons are the main reason for their enhanced sensitivity to BDNF signals (Fig. 8C/D). Above all, the proposed role for SORCS1/3 in control of TrkB activity through intracellular sorting is in perfect agreement with the established functions of related VPS10P domain receptors SORLA, sortilin, and SORCS2 in functional expression of several neurotrophin receptors (see discussion section, page 18, top).

Finally, using primary hypothalamic neurons, we now document the ability of BDNF to induce *Klf4* and, subsequently, *AgRP* transcripts, establishing AgRP as a direct downstream target of BDNF, possible through transcription factor KLF4 (Fig. 8F/G).

Taken together, we strongly believe that our previous data and our new findings in the revised manuscript provide convincing experimental support for a role of SORCS1/3 in central control of metabolism through modulation of orexigenic peptide expression. Still, we have rephrased our result and discussion sections with more caution now to indicate that yet unknown SORCS1/3-dependent mechanisms in the hypothalamus or in higher brain structures may contribute to the observed metabolic defects (page 18, top).

Point 2:

“Because SORCS1/3 are abundantly expressed in other brain regions (cortex, hippocampus), it cannot be excluded that the metabolic phenotype reported is the outcome of multiple effects originating outside of the hypothalamus.”

As elaborated under point 1 above, chronically increased expression of AgRP is a main consistent feature of SORCS1/3 deficiency seen as early as 8 weeks of age (and persisting into advanced age). Thus, this defect precedes any metabolic complications, such as adipose tissue hypertrophy or leptin resistance, suggesting NPY/AgRP neurons to be primarily impacted by loss of SORCS1 and -3 (see also our reply to point 3 of referee 2). Documented expression of SORCS1 and -3 in FACS-sorted NPY/AgRP neurons (Fig. 8E) and hypersensitivity of hypothalamic slices from S1/3 KO mice to BDNF (Fig. 8C/D) provide further experimental support for our hypothesis.

Still, we cannot fully exclude that SORCS1/3-dependent alterations in BDNF signaling in higher brain structures may contribute to the observed metabolic defects in SORCS1/3-deficient mice. Along these lines, referee 3 suggested that loss of sorting receptor expression in the hippocampus might contribute to the subtle defects in glucose handling observed in the mutant mice. This issue is now being addressed in the revised discussion section (see reply to point 1 of referee 3).

2nd Editorial Decision

9 January 2018

Thank you for the submission of your revised manuscript. We have now received the enclosed referee reports. Referee 1 was unfortunately not available to re-review the manuscript for us, and I have therefore asked referee 2 to please also assess how well referee 1's comments have been addressed.

As you will see, while the referees acknowledge that the study has been improved, referee 2 points

out that the mechanism underlying the SORCS1/3 KO phenotype remains unclear. The same concern was raised by the expert advisor I contacted before making the previous decision. I have discussed this situation with my colleagues at EMBO reports and we have decided that we can offer to publish your study if the caveats are clearly discussed in the manuscript text, and it is made clear that the proposed mechanism is one possible explanation for the phenotype but that definitive proof is missing at this point. In general, the data need to be interpreted and discussed more carefully, and over-interpretations must be avoided. Please also move figure 9 to the EV figures or the Appendix. We can offer a maximum of 5 EV figures, additional supplementary information will need to be moved to an Appendix file. You can find more information about our file types in our guide to authors online.

All other remaining referee concerns also need to be addressed and the manuscript title will need to be modified.

A few other changes will further be necessary:

- we still need the completed author checklist that can be found here:

<http://embor.embopress.org/authorguide#revision>

The checklist will also be part of the transparent peer-review process file (RPF)

- please provide up to 5 keywords for the manuscript

- please insert the ORCID IDs for the 2 corresponding authors in your profile page of our online manuscript tracking system. We can unfortunately not do this for you.

- please correct the reference style. The title needs to be added and up to 10 authors need to be listed before "et al". The EMBO reports reference style can be found in EndNote.

- remove all methods from the supplementary file. All methods need to be part of the main text. An extra supplementary file called Appendix will only be necessary if you have more than 5 EV figures.

- supplementary table 1 should be EV table 1 and needs to be uploaded as separate word or excel file.

- please use the present tense when describing your findings in the abstract

- "n" for the number of independently performed experiments is not specified in the figure legends and needs to be added. Error bars and p-values cannot be calculated if $n < 3$. The number of mice per group should also be kept.

- scale bars need to be added to all microscopy images and are missing for example in figure 4 and EV5. Both figures also need to explain what the lower panels represent.

- EMBO press papers are accompanied online by A) a short (1-2 sentences) summary of the findings and their significance, B) 2-3 bullet points highlighting key results and C) a synopsis image that is 550x200-400 pixels large (the height is variable). You can either show a model or key data in the synopsis image. Please note that text needs to be readable at the final size. Please send us this information along with the revised manuscript.

I look forward to seeing a final version of your manuscript as soon as possible. Please let me know if you have any questions or comments.

REFEREE REPORTS

Referee #2:

My previous concerns are largely addressed by new data and additional discussion, although conclusive evidence supporting the authors' model is still missing.

Minor points:

1. Figure 1E: Statistical comparisons between single mutation and double mutation are missing.
2. Figure 8E: The observation that isolated AgRP neurons contain mRNAs for SORCS1, SORCS3 and TrkB does not demonstrate that the 3 proteins are co-expressed. Each of the proteins could be expressed in a fraction of AgRP neurons.
3. The phrase "in the paraventricular nucleus of the anterior hypothalamus" (page 8) is confusing and should be revised. The anterior hypothalamus is one of hypothalamic nuclei.
4. Alpha-melanocyte-stimulating hormone shouldn't be called a neuropeptide. It is also produced in non-neuronal cells.
5. Data shown in Figure 9 could be moved to supplementary information. Conclusions shouldn't be made on the basis of the trend of data.

Referee #3:

In my opinions the authors have sufficiently addressed all of the concerns raised by all reviewers.

Additional comments from referee 2:

I think that the authors have fully addressed referee 1's concerns in points 1 and 2 by adding new data and citing literature. The observed changes in body weight, adiposity and RER of Sorcs1/3 mutant mice are physiologically significant. The authors also tried to address the referee's concerns listed in point 3 about the link between Sorcs1/3 and TrkB by performing new experiments. Data from the experiments are shown in Figures 8 and 9; however, I believe that the new data make the manuscript worse. As I commented in my review, Figure 8E is not sufficient to show that Sorcs1/3 and TrkB are expressed in the same neurons. Figure 9 only shows a trend of increasing TrkB anterograde trafficking in mutant mice, but the difference between control and mutant mice is not statistically significant. I am surprised that the authors made a conclusion on the basis of the trend. Even assuming that TrkB anterograde trafficking is increased in Sorcs1/3 mutant mice, there is not a clear link between TrkB anterograde transport and surface TrkB expression.

Overall, the manuscript reports an interesting metabolic phenotype in Sorcs1/3 mutant mice, i.e. reduced body weight, increased adiposity and reduced oxidation of lipids. The authors also show that AgRP and Npy are upregulated in the mutant mice, which could be related to an increase in surface TrkB expression in AgRP neurons. The data is consistent with the mechanism proposed by the authors, but they are just some correlation data. The authors did not perform any experiments to show that the change in AgRP neurons is responsible for the observed metabolic phenotype. Changes in other neuronal populations, such as POMC neurons, could lead to the observed phenotype. If you accept the manuscript for publication, I think that Figure 9 should be removed from the manuscript, at least from main figures.

2nd Revision - authors' response

15 January 2018

COMMENTS BY THE EDITOR

Point 1) *“Caveats are clearly discussed in the manuscript text, and it is made clear that the proposed mechanism is one possible explanation for the phenotype but that definitive proof is missing at this point.”*

Our manuscript now clearly stresses the caveats in interpretation of our data.

- 1) It specifically states that expression of SORCS1 and -3 in other brain areas than the hypothalamus may contribute to the metabolic phenotype (page 19, line 3 from top).
- 2) We now refrain from citing the live cell imaging data (former Fig. 9A-D) as experimental evidence for a role of SORCS1/3 in sorting of TrkB along neurites as these data showed only a trend ($p=0.1$). Also, we removed these data from the main figure (see point 2 below).
- 3) As for our working model, we now solely discuss a role for sorting receptors SORCS1 and -3 in control of subcellular localization and cell surface exposure of TrkB, a hypothesis convincingly supported by statistically significant results from our

proteomics and brain subcellular fractionation experiments (Fig. 7A-E).

4) Most importantly, we now end the discussion section with a clear statement that multiple scenarios for SORCS1/3 action may be envisioned and that definitive proof is still missing. The respective text reads: “*Clearly, different scenarios may be envisioned whereby these multifunctional receptors impact energy homeostasis. Our data suggest a model in which SORCS1 and -3 functionally interact with TrkB to alter subcellular localization and, thereby, reduce responsiveness of this receptor to BDNF signals in hypothalamic neurons. Obviously, further studies will be required to ultimately resolve the molecular mechanism(s) underlying the action of these sorting receptors in regulation of energy balance.*”

Point 2) “*Please also move figure 9 to the EV figure. We can offer a maximum of 5 EV figures, additional supplementary information will need to be moved to an Appendix file.*” Also, supplementary table 1 should be EV table 1 and needs to be uploaded as separate word or excel file.”

As requested by you (and by referee #2), we now moved figure 9 to the EV figure section (EV figure 4). Also, the supplementary table is now designated as EV table 1 and uploaded as word file.

To conform with a maximum of 5 EV display items, we moved three EV figures to an appendix section uploaded as pdf. All in all, our manuscript now includes 8 main figures, 5 EV display items, and 3 supplementary figures in the appendix.

Point 3: “*We still need the completed author checklist*”

The author checklist has been completed and uploaded.

Point 4: Additional editorial comments

- The title has now been changed to conform with the length requirements.
- Five keywords have been included in the manuscript text.
- ORCID IDs for the two corresponding authors have been added to the author profile pages.
- The reference style has now been corrected.
- All methods relevant for main and EV figure data have been moved to the main text. Only methods relevant for the supplementary figures remained in the appendix.
- Present tense is used in the abstract throughout.
- Where applicable, "n" for the number of independently performed experiments or number of mice per genotype group are specified in the figure legends.
- No error bars and p-values were calculated if $n < 3$ in the previous version of the manuscript. The number of mice per group is stated in all figure legends.
- Missing scale bars were added to figure 4 and EV5 (now Appendix figure S2). Also, explanations are given to explain what the lower panels represent.

Point 5: *Summary, bullet points, and synopsis image.*

We included a short summary and bullet points in the manuscript file (page 2). Also, we uploaded a synopsis image. As discussed under point 1 above, in the synopsis image we now display roles for SORCS1 and -3 in control of subcellular localization of TrkB (not trafficking along neurites).

Additional points from Referee #2:

Point 1. “*Figure 1E: Statistical comparisons between single mutation and double mutation are missing.*”

The information about statistical comparisons between genotypes is now given in the figure legend.

Point 2. “*Figure 8E: The observation that isolated AgRP neurons contain mRNAs for SORCS1, SORCS3 and TrkB does not demonstrate that the 3 proteins are co-expressed. Each of the proteins could be expressed in a fraction of AgRP neurons.*”

We now have modified the text in result and discussion sections to state that SORCS1/-3 and TrkB all are expressed in AgRP neurons (not co-expressed in AgRP neurons).

Point 3. “*The phrase "in the paraventricular nucleus of the anterior hypothalamus" (page 8)*

is confusing and should be revised. The anterior hypothalamus is one of hypothalamic nuclei.”

As suggested by the reviewer, this phrase has now been corrected.

Point 4. *“Alpha-melanocyte-stimulating hormone shouldn't be called a neuropeptide. It is also produced in non-neuronal cells.”*

We have changed the wording to hormone (not neuropeptide).

Point 5. *“Data shown in Figure 9 could be moved to supplementary information. Conclusions shouldn't be made on the basis of the trend of data.”*

As detailed in our response to point 1 of the editor, the live cell imaging data have now been moved to EV figure 4. Also, we now refrain from specifically stating a role for SORCS1/-3 in anterograde/retrograde sorting of TrkB. Also, our working model described in the discussion section, and shown in the synopsis image, does not suggest a role for SORCS1/-3 in trafficking along neurites. Rather, we propose a function for these receptors in modulating the subcellular localization and cell surface exposure of TrkB (as documented by our proteomics and subcellular fractionation studies in Fig. 7).

ADDITIONAL COMMENTS FROM REFEREE 2 TO OUR RESPONSE TO REFEREE 1:

Point 6: *“As I commented in my review, Figure 8E is not sufficient to show that Sorcs1/3 and TrkB are expressed in the same neurons.”*

“Figure 9 only shows a trend of increasing TrkB anterograde trafficking in mutant mice, but the difference between control and mutant mice is not statistically significant. Even assuming that TrkB anterograde trafficking is increased in Sorcs1/3 mutant mice, there is not a clear link between TrkB anterograde transport and surface TrkB expression.”

The wording in our results section has been changed to read expression (not co-expression) of the three proteins in AgRP neurons (see point 2 above).

As for the live cell imaging data (Fig. 9), we agree with the comments made by this reviewer and by the editor and we have moved these data to EV figure 4. Also, we now propose a role for SORCS1/-3 in surface exposure (not anterograde transport) of TrkB.

Point 7: *“The authors did not perform any experiments to show that the change in AgRP neurons is responsible for the observed metabolic phenotype. If you accept the manuscript for publication, I think that Figure 9 should be removed from the manuscript, at least from main figures.”*

As requested, we have moved the data from previous figure 9 to EV figure 4. As discussed under point 1 of the editor above, we also now phrase our working model with even more caution, and we clearly address the caveats and the lack of ultimate proof for our model in the final paragraph of the discussion section.

Corresponding Author Name: Thomas Willnow, Aygul Subkhangulova

Manuscript Number: EMBOR-2017-44810V2